



# Recent Advances in Aerosol Optical Depth Measurements in Polar Regions: Insights from the Polar-AOD Program

Simone Pulimeno[1,2], Angelo Lupi[2], Vito Vitale[2], Claudia Frangipani[2], Carlos Toledano[3],
Stelios Kazadzis[4], Natalia Kouremeti[4], Christoph Ritter[5], Sandra Graßl[5], Kerstin Stebel[6],
Vitali Fioletov[7], Ihab Abboud[7], Sandra Blindheim[8], Lynn Ma[9], Norm O'Neill[10], Piotr Sobolewski[11],
Pawan Gupta[12], Elena Lind[12], Thomas F. Eck[12], Antti Hyvärinen[13], Veijo Aaltonen[13], Rigel Kivi[14],
Janae Csavina[15], Dmitry Kabanov[16], Sergey M. Sakerin[16], Olga R. Sidorova[17], Robert S. Stone[18,*],
Hagen Telg[18], Laura Riihimaki[18], Raul R. Cordero[19,20], Martin Radenz[21], Ronny Engelmann[21],
Michel Van Roozendal[22], Anatoli Chaikovsky[23], Philippe Goloub[24], Junji Hisamitsu[25], and
Mauro Mazzola[2]

[1]Ca' Foscari University of Venice, Department of Environmental Sciences, Informatics and Statistics, 30123 Venice, Italy
[2]National Research Council, Institute of Polar Sciences, 40129 Bologna, Italy
[3]University of Valladolid, 47002 Valladolid, Spain
[4]Physikalisch-Meteorologisches Observatorium Davos, 7260 Davos, Switzerland
[5]Alfred Wegener Institute for Polar and Marine Research, 14473 Potsdam, Germany
[6]NILU, 2007 Kjeller, Norway
[7]Environment and Climate Change Canada, M3H 5T4 Ontario, Canada
[8]Andøya Rocket Range, 8480 Andenes, Norway
[9]Environmental Science and Technologies, Brookhaven National Laboratory, 11973 New York, USA
[10]Sherbrooke University, QC J1N 3C6 Quebec, Canada
[11]Polish Academy of Sciences, Institute of Geophysics, 01-452 Warsaw, Poland
[12]NASA Goddard Space Flight Center, 20771 Greenbelt, USA
[13]Finnish Meteorological Institute, 00101 Helsinki, Finland
[14]Finnish Meteorological Institute, 99600 Sodankylä, Finland
[15]National Ecological Observatory Network, 80301 Boulder, USA
[16]Zuev Institute of Atmospheric Optics, Russian Academy of Science, 634055 Tomsk, Russia
[17]Arctic and Antarctic Research Institute, 199397 St Petersburg, Russia
[18]National Oceanic and Atmospheric Administration, 20230 Washington, USA
[19]University of Groningen, 8911 CE Leeuwarden, The Netherlands
[20]Santiago de Chile University, 9170022 Santiago, Chile
[21]Leibniz Institute for Tropospheric Research, 04318 Leipzig, Germany
[22]Belgian Institute for Space Aeronomy, 1180 Brussels, Belgium
[23]Institute of Physics, National Academy of Sciences of Belarus, 220072 Minsk, Belarus
[24]University of Lille, Atmospheric Optics Laboratory, 59655 Villeneuve d'Ascq Cedex, France
[25]Japan Meteorological Agency, 105-8431 Tokyo, Japan
[*]retired

**Correspondence:** Simone Pulimeno (simone.pulimeno@unive.it)

**Abstract.**

A multi-year analysis of aerosol optical depth (AOD, $\tau$) and Ångström exponent ($\alpha$) was conducted using ground-based photometer data from 15 Arctic and 11 Antarctic sites. Extending the dataset of Tomasi et al. (2015) through December 2024,





the study incorporates stellar and lunar photometric observations to fill data gaps during the polar night. Daily mean values of
$\tau$ at 0.500 $\mu$m and $\alpha$ (0.440–0.870 $\mu$m) were used to derive monthly means and seasonal histograms.

In the Arctic, persistent haze events in winter and early spring lead to peak $\tau$ values. A decreasing trend in Arctic $\tau$ suggests the impact of European emission regulations, while biomass-burning aerosols are becoming more significant. In Antarctica, $\tau$ increases from the plateau to the coast. Fine-mode aerosols dominate in summer-autumn, while coarse-mode particles are more prevalent in winter-spring. Shipborne photometer data align well with ground-based measurements, confirming the reliability
of mobile observations.

Trend analyses using the Mann-Kendall test and Theil-Sen regression indicate a significant negative trend in $\tau$ at Andenes (-2.43% per year), likely driven by reduced anthropogenic emissions. Antarctic stations such as Syowa and South Pole show positive trends (+3.84% and +3.54% per year), though these are subject to uncertainties from data limitations and instrument changes.

This work contributes to the Polar-AOD network (https://polaraod.net/; last access 15/05/2025), enhancing the understanding of aerosol variability and long-term trends in polar regions while promoting open data access for the scientific community.

## 1  Introduction

Atmospheric aerosols play a crucial role in the Earth's atmosphere and represent some of its most dynamic components. Since the pre-industrial era, anthropogenic activities have significantly increased the concentration of atmospheric aerosols, partic-
ularly sulfate and carbonaceous aerosols. This rise has influenced the absorption and scattering of incoming solar radiation, thereby affecting both the microphysical and macrophysical properties of clouds, as well as their radiative properties. Although there is a high degree of uncertainty regarding the role of aerosols in the climate system, they are mostly associated with a negative feedback mechanism, i.e. aerosols interact with solar radiation, resulting in a net cooling effect (Intergovernmental Panel on Climate Change (IPCC) (2023)).

The Arctic atmosphere is highly stratified, with frequent inversions near the surface. This stability reduces turbulence and dry deposition at the surface. Surfaces of constant potential temperature form a dome over the Arctic, creating a transport barrier that isolates this environment from the rest of the atmosphere (Klonecki et al. (2003)). When addressing pollution levels in the Arctic, it is important to consider the variation in pollution transport due to the North Atlantic Oscillation (NAO). During the positive phases of the NAO, transport patterns from the northern mid-latitude continents (Europe, North America, and Asia) are
significantly enhanced (Stohl (2006)). The Arctic Haze phenomenon significantly impact the Arctic region. Since the 1950s, pilots have documented a reduction in visibility due to the presence of haze (Raatz et al. (1985)). Scientists debated the origin of the haze until the 1970s, when it was proposed that Arctic haze originated not only from natural sources (Warneke et al. (2009)) but also from anthropogenic emissions in the northern mid-latitudes. During winter, the expansion of the Polar dome (down to 40° N) facilitates the intrusion and the transport of pollutants into the Arctic atmosphere over thousands of kilometers. The
inefficiency of removal mechanisms during winter and early spring contributes to the seasonality of this phenomenon (Shaw (1995)).



Polar aerosols originate from both natural and anthropogenic sources. In the Arctic region, the majority of the aerosol mass fraction consists of oceanic sea-salt, mineral dust, non-sea-salt sulfate, and biomass burning combustion products (Tomasi et al. (2015)). Conversely, anthropogenic aerosols have a different composition, characterized by high concentrations of black

carbon (BC), sulfates, and nitrates, which are typical signatures of traffic and industrial emissions (Quinn et al. (2007); Sharma et al. (2006)).

In the Southern Hemisphere, the atmosphere is stably stratified much like in the Arctic, though katabatic winds carry air from the interior plateau to the coasts. Key transport processes in this region are similar to those in the Arctic and include: (i) lifting at the Antarctic front, (ii) lifting at lower latitudes, and (iii) descent due to radiative cooling of upper-tropospheric air masses

(Tomasi et al. (2007)). In Antarctica, aerosols at coastal sites are almost totally from natural processes, with high percentages of sea-salt mass content, non-sea-salt sulphate, and mineral dust (Tomasi et al. (2012)). In this region, only a small fraction of the total aerosol mass is of anthropogenic origin, such as nitrates and BC (Cordero et al. (2022)). Significant aerosol sources in both the Arctic and Antarctic regions include commercial ships, primarily operating in the Northern Hemisphere along the Northern Sea Route and the Transpolar Sea Route, and cruise ships mainly in Antarctica. Additionally, diesel generators used

for energy production, aircraft emissions, and access to strategic resources (such as ore and oil) can also be considered as important sources of pollution at both poles.

The aim of this paper is to update the aerosol climatologies at both poles, developing the work published by Tomasi et al. (2015). This study was limited by the technology available at the time, relying solely on the solar photometry technique. This posed a significant limitation for scientists working in polar regions, where the Sun is absent for several months during the

polar night. For example, at the Arctic station of Ny-Ålesund (Svalbard, Norway), the Sun remains below 5° of elevation from October 10th to March 4th, severely restricting the period available for conventional photometric measurements. To overcome this challenge, Lidar instruments can be used, as they provide aerosol extinction profiles even at night, which can then be converted to optical depth for comparison with photometric data. However, there are too few Lidar systems in polar regions to provide a comprehensive image of these environments. The lunar photometry technique, which has developed over the last

decade, has proven to be a suitable technique in polar areas, where $\tau$ values are often below 0.05 (Mazzola et al. (2024)). To address this issue and fill historical gaps in $\tau$ climatology, it has been proposed to use solar and lunar photometry techniques in synergy. AERONET stations at both poles with measurements up to December 2024, have been selected for this work (AERONET: https://aeronet.gsfc.nasa.gov/; last access 15/05/2025).

The paper is organized as follows. In the next section, a description of the main characteristics of both solar and lunar

photometry techniques is provided. Section 3 presents the main optical characteristics of polar aerosols in the Arctic region, while Section 4 focuses on measurements in Antarctica.



## 2 Ground-based remote sensing measurements

Remote sensing ground-based techniques are commonly used to study the characteristics of the atmospheric column. Specifically, photometry has proven to be effective also in polar areas, where background values are smaller compared to continental areas due to a cleaner atmosphere (Mazzola et al. (2012)).

A sun-photometer is an instrument that is kept oriented towards the Sun to detect solar radiation attenuated by particles in the atmospheric column along the slant path from the top of the atmosphere (TOA) to the ground. The more particles present in the atmospheric column, the more attenuated the direct solar radiation detected by the photometer will be. This attenuation depends on the aerosol optical depth (AOD), represented by the symbol $\tau_{(\lambda)}$, which is the integral of the volume aerosol extinction coefficient along the vertical path of the atmosphere (Tomasi et al. (2015)).

In recent decades, several sun-photometer models have been developed and implemented in major photometry networks worldwide. The most important of these networks are: (i) AERONET (AErosol RObotic NETwork), established by NASA, which has provided long-term measurements for over 25 years with standardized calibration, processing, and distribution processes (Holben et al. (1998)); (ii) SKYNET, initiated under the WCRP/GAME, which evaluates long-term variations in aerosol concentrations and is mainly distributed in Asia (Nakajima et al. (2020)); (iii) GAW-PFR, an international network that measures AOD at GAW stations (Kazadzis et al. (2018)).

We focused primarily on AERONET sites in both the Arctic and Antarctica, as this network provides highly accurate AOD measurements (accuracy of 0.01 for the visible and NIR wavelengths at optical airmass of 1) and has the widest coverage at the poles. The high latitudes of these sites result in large airmass values, typically in the range of 2 to 7. This leads to a reduction in AOD calibration uncertainty by a factor of $1/m$, where $m$ is the airmass (Eck et al. (1999)). In 2018, the latest AERONET Version 3 (V3) algorithm was published, featuring a fully automatic cloud screening procedure and instrument anomaly controls (Giles et al. (2019)). AERONET includes a component dedicated to ship-borne AOD measurements using the manual sun photometer Microtops II. Since 2004, this instrument has been routinely deployed on research vessels to monitor aerosol properties over the oceans. The Maritime Aerosol Network (MAN) enables the study and evaluation of aerosol properties across various oceanic regions, including polar sectors.

In addition to CIMEL CE318 data from AERONET, other international research groups provided AOD measurements obtained using various photometer models, including the SPM multiwavelength sun photometer and its simplified version the SP-9 (Sakerin et al. (2013)); the Carter-Scott Design Middleton SP02 sun photometer (McArthur (2005)); the SP1A developed by Dr. Schults and Partner GmbH (Stock et al. (2014)); the Precision Filter Radiometer (PFR) (Wehrli (2000)); and the MS110 sun photometer (Kim et al. (2005)).

### 2.1 Solar photometry

The fundamental equation used in sun-photometry to retrieve AOD is the Lambert-Beer law (Shaw (1976)). Specifically, this equation is applied to the raw signal at a given wavelength $\lambda$ ($V_{(\lambda)}$) measured by the instrument at ground level, and to the



signal the photometer would detect at the TOA ($V_{0(\lambda)}$):

$$V_\lambda = \frac{V_{0,\lambda}}{R^2} e^{-\tau_{TOD,\lambda} m} \tag{1}$$

where $R$ represents the Earth-Sun distance in AU, $m$ is the optical air mass that indicates the relation between extinction in the vertical column and that in the measurements slant path (which is related to the zenith angle of the target) and $\tau_{TOD,\lambda}$ is the total optical depth (TOD).

In order to obtain reliable values of $V_{0,\lambda}$ at a given wavelength, the Langley plot method can be applied (Shaw (1983)). This method involves applying a linear regression between the logarithm of the signals measured by the instrument ($\ln V_\lambda$) and the calculated values of air masses ($m$). The intercept of this line represents the value of $V_{0,\lambda}$ at the TOA, and the slope represents the $\tau_{TOD,\lambda}$. Only the signals measured at different spectral channels within an air mass range usually between 2 and 5 are considered. Values of $m<2$ are not used because the rate of change of air mass is very small, with a higher likelihood that changing weather conditions will influence the regression. Conversely, values of $m>5$ are discarded due to greater uncertainty in the value of $m$ itself caused by corrections due to the phenomenon of refraction. To avoid errors in corrections at high solar zenith angles, the range normally used in the Langley plot method is $m$ between 2 and 5 (Alexandrov et al. (2004); Mazzola et al. (2010)).

In the term $\tau_{TOD,\lambda}$, the contribution due to scattering and absorption by gases is included, so that Eq.(1) can be rewritten in logarithmic form as:

$$\ln V_\lambda = \ln(V_{0,\lambda} R^{-2}) - (\tau_{a,\lambda} m_a + \tau_{R,\lambda} m_R + \tau_{g,\lambda} m_g) \tag{2}$$

The subscript 'a' stands for aerosol, 'R' for Rayleigh scattering by molecules, and 'g' for absorption gases. At this point, the AOD can de directly derived by Eq.(2) as:

$$\tau_{a,\lambda} = -\frac{1}{m_a}\left[\ln\left(\frac{V_\lambda}{V_{0,\lambda} R^2}\right) - \tau_{R,\lambda} m_R - \tau_{g,\lambda} m_g\right] \tag{3}$$

Since the vertical distribution is different for any gas, several air mass factors are taken into account; for example, ozone is mainly stratospheric, while carbon dioxide is uniformly mixed (González et al. (2020)). The gaseous species considered in the term $\tau_{g,\lambda} m_g$ of Eq.(3) are ozone $O_3$, nitrogen dioxide $NO_2$, water $H_2O$, carbon dioxide $CO_2$, and methane $CH_4$.

Another important parameter that can be estimated from $\tau_\lambda$ measurements is the Ångström exponent ($\alpha$) calculated as follows:

$$\alpha = -\frac{log\left(\frac{\tau_{\lambda_1}}{\tau_{\lambda_2}}\right)}{log\left(\frac{\lambda_1}{\lambda_2}\right)} \tag{4}$$

where $\tau_{\lambda_1}$ and $\tau_{\lambda_2}$ are the AOD values at the wavelengths $\lambda_1$ and $\lambda_2$, usually 0.440 and 0.870 $\mu m$. This parameter quantifies the wavelength dependence of AOD, providing insight into the size distribution of atmospheric particles (Kaskaoutis et al. (2007)).

While $\tau_\lambda$ provides information about the extinction caused by the presence of aerosol particles along the vertical atmospheric path, $\alpha$ reflects the contributions of different particle sizes to this extinction. Values of $\alpha$ greater than 1.3 are typically associated





with a predominance of very fine particles, whereas values of $\alpha$ less than 1.0 indicate the presence of particles in accumulation and coarse mode, which produce a larger extinction effect (Iqbal (1983)). Additionally, the $\alpha$ value in the 0.440-0.870 $\mu m$ range is dominated by the coarse mode (Eck et al. (2010)).

## 2.2 Lunar photometry

In the last decades, many attempts have been made to use the Moon as a light source to retrieve aerosol properties. The stability
of the lunar surface reflectance makes the Moon a nearly perfect calibration source. However, there are significant challenges due to the non-uniformity of the lunar surface albedo resulting from the presence of lunar maria and highlands, the brightness variation arising from lunar phase and libration, the strong dependence of surface reflectivity on phase angle, and the fact that cloud cover can block or reduce the Moon's irradiance. The complexity of these dependencies effectively mandates the use of a lunar radiometric model to compare against spacecraft observations of the Moon. The USGS in Flagstaff (Arizona,
US) has acquired the observational data and proposed the RObotic Lunar Observatory (ROLO) model (Kieffer and Stone (2005)). This model can provide the exoatmospheric lunar irradiance for any given location and time. The model is based on fitting thousands of lunar measurements acquired over more than 8 years with the ground-based ROLO telescopes in 32 wavelength bands from 0.350 to 2.450 $\mu m$. The ROLO model uses an empirically derived analytic equation to predict the lunar disk-equivalent reflectance ($A_k$) in the spectral band k using only geometric variables (Kieffer and Stone (2005)):

$$145 \quad lnA_k = \sum_{i=0}^{3} a_{ik}g^i + \sum_{j=1}^{3} b_{jk}\Phi^{2j-1} + c_1\phi + c_2\theta + c_3\Phi\phi + c_4\Phi\theta + d_{1k}e^{-g/p_1} + d_{2k}e^{-g/p_2} + d_{3k}cos((g-p_3)/p_4) \quad (5)$$

Where $g$ is the absolute phase angle, $\theta$ and $\phi$ are the selenographic latitude and longitude of the observer, and $\Phi$ is the selenographic longitude of the Sun. The ROLO model provides exo-atmospheric lunar irradiance with relatively high precision. The band-average absolute residuals are about 1%, based on comparisons between ROLO empirical irradiances and hundreds of ROLO observations. This high precision makes the ROLO model a valuable tool for calibrating measurements and interpreting
data for aerosol property retrievals. As always, for the retrieve of AOD during nighttime, the Beer-Lambert law can be used:

$$V_\lambda = V_{0,\lambda}e^{-m(\theta)\tau_\lambda} \quad (6)$$

Where $V_\lambda$ is the output voltage, $V_{0,\lambda}$ the extraterrestrial voltage which include lunar phase variations as well as Earth-Moon and Moon-Sun distances, $m$ is the relative optical air mass (and function of the moon zenith angle $\theta$), and $\tau_\lambda$ the spectral optical depth. To account for the change in lunar illumination during the course of the night, and for the distant effect on lunar
irradiance, the $V_0$ term of Eq.(6) can be modified as:

$$V_{0,j} = I_{0,j}k_j \quad (7)$$

Where $I_{0,j}$ is the extraterrestrial irradiance in a certain channel with a central wavelength at $j$, and $k_j$ is a constant that depends on the instrument features such as the calibration coefficient and the instrument's solid angle-of-view. $I_{0,j}$ is calculated using the ROLO lunar disk-equivalent reflectances ($A_k$) in Eq.(5). The exact formula can be found in Barreto et al. (2013). In the





same paper, authots proposed the Lunar-Langley Method for the calibration of the instrument. Basically, the logarithmic form of Eq.(6) and Eq.(7), together with a least square fitting procedure are used to obtain the instrument's calibration constant ($k_j$) as the intercept of the fitting line. Once these constants are known, it is possible to retrieve AOD from an individual measurement:

$$\tau_{a,j} = \frac{\ln k_j - \ln\left(\frac{V_j}{I_{0,j}}\right) - m_{atm}(\theta)\tau_{atm,j}}{m_{a(\theta)}} \tag{8}$$

The subscript 'atm' accounts for air mass and optical depth of each atmospheric attenuator with the exception of aerosols. Román et al. (2020), proposed the use of the RIMO (ROLO Implementation for Moon's Observation) model to retrieve AOD during night-time, based on the assumption that the calibration constants for solar channels can be transferred to the Moon. Because authors found an underestimation of AODs retrieved by using this model (dependent on the optical air mass), they proposed a correction factor that, multiplied by the RIMO value, gives a more accurate extraterrestrial lunar irradiance that can be used for a more accurate retrieval of AODs during night.

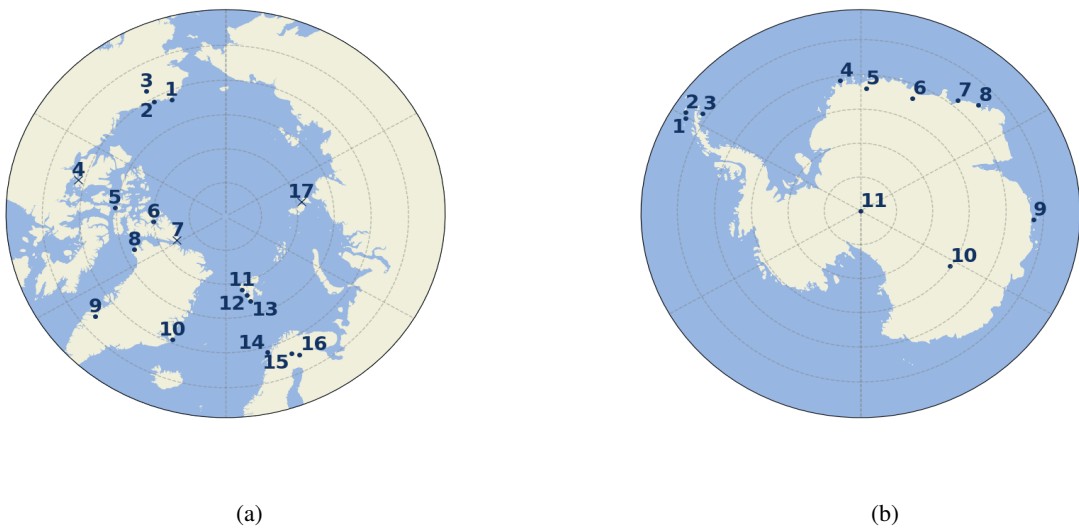

(a)                                              (b)

**Figure 1.** Part (a) map of the Arctic showing the geographical position of the stations labeled as follows: (1) Barrow, (2) Oliktok, (3) Toolik Lake, (4) Cambridge Bay, (5) Resolute Bay, (6) Eureka (OPAL), (7) Alert, (8) Thule, (9) Kangerlussuaq, (10) Ittoqqortoormiit, (11) Ny-Ålesund, (12) Barentsburg, (13) Hornsund, (14) Andenes, (15) Matorova, (16) Sodankylä, and (17) Cape Baranova. Part (b) presents a map for the Antarctic stations, with the locations marked as: (1) Juan Carlos I, (2) Escudero, (3) Marambio, (4) Neumayer, (5) Troll/Trollhaugen, (6) Utsteinen, (7) Syowa, (8) Vechernaya Hill, (9) Mirny, (10) DomeC and (11) South Pole. Stations marked with 'X' have been studied but not shown in this paper.



| Stations - ID | Managing Institutions | Instrument model | Coordinates and Altitude | Measurement Period | Lunar Data started |
|---|---|---|---|---|---|
| **Arctic sites** | | | | | |
| Alert (Canada) - 7 | Environment and Climate Change Canada, Canada | CE318 | 82.4501 N, 62.5074 W, 210 m a.m.s.l. | March 2024 - November 2024 [11,641] | October 2024 - November 2024 [811] |
| Andenes (Norway) - 14 | Andøya Rocket Range, Norway - University of Valladolid, Spain | CE318 | 69.2783 N, 16.0086 E, 379 m a.m.s.l. | February 2002 - December 2023 [55,719] | December 2016 [8,046] |
| Barentsburg (Norway) - 12 | Arctic and Antarctic Research Institute - Zuev Institute of Atmospheric Optics, Russia | SPM and SP-9 | 78.0590 N, 14.2192 E, 7 m a.m.s.l. | March 2011 - August 2023 [605] | no |
| Barrow (US) - 1 | Atmospheric Radiation Measurements (ARM), US | CE318 | 71.3226 N, 156.6151 W, 8 m a.m.s.l. | July 1997 - December 2023 [41,003] | October 2018 [2,011] |
| Barrow (US) - 1 | NOAA, US | SP02 | 71.1926 N, 156.3615 W, 8 m a.m.s.l. | March 2001 - October 2016 [1,764] | no |
| Cambridge Bay (Canada) - 4 | Environment and Climate Change Canada, Canada - University of Sherbrook, Canada | CE318 | 69.1213 N, 105.0394 W, 5 m a.m.s.l. | September 2024 - October 2024 [110] | September 2024 [16] |
| Cape Baranova (Russia) - 17 | Zuev Institute of Atmospheric Optics, Russia | SPM | 79.1682 N, 101.3705 E, 20 m a.m.s.l. | April 2018 - August 2021 [59] | no |
| Eureka OPAL (Canada) - 6 | Environment and Climate Change Canada, Canada | CE318 | 79.9902 N, 85.9391 W, 5 m a.m.s.l. | April 2007 - December 2024 [95,421] | April 2016 [6,044] |
| Hornsund (Norway) - 13 | Polish Academy of Sciences, Poland | CE318 | 77.0014 N, 15.5402 E, 12 m a.m.s.l. | May 2004 - December 2023 [30,382] | October 2021 [113] |

*Continued on next page*



| | | | Arctic sites (continued) | | |
|---|---|---|---|---|---|
| **Stations - ID** | **Managing Institutions** | **Instrument model** | **Coordinates and Altitude** | **Measurement Period** | **Lunar Data started** |
| Ittoqqortoormiit (Greenland) - 10 | NASA Goddard Space Flight Center, US | CE318 | 70.4848 N, 21.9512 W, 68 m a.m.s.l. | October 2009 - November 2022 [38,943] | September 2020 [667] |
| Kangerlussuaq (Greenland) - 9 | NASA Goddard Space Flight Center, US | CE318 | 66.9958 N, 50.6214 W, 320 m a.m.s.l. | April 2008 - December 2023 [68,634] | March 2017 [6,826] |
| Matorova (Finland) - 15 | Finnish Meteorological Institute, Finland | CE318 | 67.9999 N, 24.2400 E, 340 m a.m.s.l. | September 2020 - September 2024 [32,116] | October 2020 [6,354] |
| Ny-Ålesund (Norway) - 11 | Alfred Wegener Institute, Germany - University of Valladolid, Spain | CE318 | 78.9232 N, 11.9230 E, 7 m a.m.s.l. | June 2017 - December 2023 [26,426] | October 2017 [2,144] |
| Ny-Ålesund (Norway) - 11 | Alfred Wegener Institute, Germany | SP1A | 78.9233 N, 11.9292 E, 17 m a.m.s.l. | March 2012 - September 2023 [407,945] | no |
| Ny-Ålesund (Norway) - 11 | Physical Meteorological Observatory Davos, Switzerland - NILU, Norway | PFR | 78.9232 N, 11.9293 E, 15 m a.m.s.l. | May 2005 - September 2023 [588,041] | no |
| Oliktok (US) - 2 | Atmospheric Radiation Measurement (ARM), US | CE318 | 70.4995 N, 149.8800 W, 2 m a.m.s.l. | September 2013 - June 2021 [20,193] | September 2018 [1,279] |
| Resolute Bay (Canada) - 5 | Environment and Climate Change Canada, Canada - AEROCAN, Canada | CE318 | 74.7051 N, 94.9694 W, 35 m a.m.s.l. | July 2004 - December 2024 [28,391] | no |
| Sodankylä (Finland) - 16 | Finnish Meteorological Institute, Finland | CE318 | 67.3666 N, 26.6295 E, 184 m a.m.s.l. | March 2013 - December 2023 [21,500] | November 2017 [4,062] |
| Thule (Greenland) - 8 | NASA Goddard Space Flight Center, US | CE318 | 76.5145 N, 68.7431 W, 225 m a.m.s.l. | March 2007 - December 2023 [73,819] | May 2017 [3,744] |





| Arctic sites (continued) | | | | | |
|---|---|---|---|---|---|
| **Stations - ID** | **Managing Institutions** | **Instrument model** | **Coordinates and Altitude** | **Measurement Period** | **Lunar Data started** |
| Toolik Lake (US) - 3 | National Ecological Observatory Network, US | CE318 | 68.6610 N, 149.3704 W, 843 m a.m.s.l. | February 2017 - December 2023 [14,902] | no |

Table 1: List of Arctic stations using different models of photometer. For each station, the coordinates and altitude are specified, along with the measurement period for solar photometry and the installation period for the lunar model. The number of measurements for both the solar and lunar periods is provided in parentheses.

## 3 Measurements in the Arctic

To conduct an in-depth analysis of polar aerosol optical characteristics, sun- and moon-photometer measurements of $\tau$ across various spectral channels in the visible and near-infrared can be examined to evaluate the Ångström exponent $\alpha$, as described by Eq. (4).

Table 1 presents information on the 15 Arctic sites analyzed in this paper, where direct radiation measurements were conducted under partly cloudy sky conditions over the past decades. The geographical locations of these sites are shown in Fig. 1a for the Arctic region.

Several international institutions provided high-quality measurements of $\tau$ and $\alpha$ using different photometer models. The specific wavelengths used by these photometers, along with details about calibration and cloud-screening procedures, are described singularly. For what concerns AERONET stations, level 2 data for solar photometry and level 1.5 data for lunar photometry have been used (as level 2 data for lunar photometry are not yet available on AERONET). Both datasets are of high quality, with a slight distinction: near-real-time level 1.5 data have been corrected for the presence of clouds during measurements, whereas level 2 data have undergone additional quality controls with pre- and post-calibration procedures applied (Giles et al. (2019)). Furthermore, the cloud screening at night does not include lunar aureole data due to insufficient measurement signals and therefore the aureole radiance curvature test for this cirrus clouds could not be applied at night (Giles et al. (2019)).

Since not only AERONET data but also measurements from several other photometers were used, the initial datasets were characterized by different time intervals. To account for these differences, individual measurements of $\tau_\lambda$ recorded under cloud-free sky conditions at each station were first averaged on an hourly basis. These hourly averaged values were then used to calculate multi-year monthly mean values of $\tau(0.500\mu m)$ and $\alpha$. When analyzing these parameters, it is important to remember that AOD data are not normally distributed, especially when episodic or extreme events occur. In such cases,





the mean value can be heavily influenced by these episodes, as well as by the overall data availability for a specific day or month. For example, a month with two episodic high-AOD events will show a higher monthly mean if there are many cloudy

days (resulting in fewer AOD measurements) compared to a month with more frequent clear-sky observations and the same number of episodic events. Additionally, since AOD measurements depend strongly on weather conditions, the data series are often discontinuous, with large gaps during periods of persistent cloud cover, polar night, or instrument downtime. These discontinuities must be carefully considered when interpreting long-term trends or variability.

We also defined relative frequency histograms for both parameters during the following seasons: winter (December to Febru-

ary), when intrusions of polluted particles into the Arctic atmosphere are more frequent; spring (March to May); summer (June to August), to characterize background aerosols; and autumn (September to November).

## 3.1 Northern America

Several multi-year sets of sun-photometer measurements have been collected from coastal stations on the Arctic Ocean, as well as from a more continental sites in Northern America.

At Barrow, the Atmospheric Radiation Measurement (ARM) program conducted observations using a CIMEL sun photometer from July 1997 to December 2024. In 2018, this instrument was replaced with a sun-sky-lunar CIMEL photometer, which also began providing lunar data starting in October 2018. Additionally, the National Oceanic and Atmospheric Administration (NOAA) studied aerosol extinction properties at the same site using a Carter Scott SP02 sun photometer from March 2001 to October 2016. The SP02 measured solar radiation at four spectral channels, centered at wavelengths of 0.412, 0.500, 0.675,

and 0.862 $\mu m$. Consistent with AERONET stations, $\tau$ was analyzed at 0.500 $\mu m$, while the $\alpha$ was studied over the spectral range of 0.412–0.862 $\mu m$. At Barrow (Fig.2), the monthly mean values of $\tau(0.500 \mu m)$ measured by the CIMEL increased from approximately 0.07 in February to 0.10 in July, before decreasing to 0.04 in October. The standard deviation ($\sigma$) of these measurements exceeded 0.17 in June and July but remained below 0.04 in other months. Regarding the $\alpha$, Barrow exhibited typical Arctic site behavior. Specifically, $\alpha$ was lower during the winter and early spring, coinciding with the period of most

intense Arctic haze and elevated atmospheric pollution; low $\alpha$ values can also result from the presence of wind-blown sea salt (O'Neill et al. (2016)), or thin cirrus clouds. During the polar spring, coarse-mode Asian dust could be mistaken for Arctic haze (AboEl-Fetouh et al. (2020)) in the absence of lidar measurements. Conversely, $\alpha$ was higher during the summer and autumn, when the atmosphere is generally cleaner, and the aerosol population is dominated by smaller particles. The mean $\alpha$ values recorded by the two instruments were similar. For winter-spring, $\alpha$ was 1.00 and 1.09, as measured by the CIMEL and

SP02, respectively. For summer-autumn, $\alpha$ values were 1.45 for the CIMEL and 1.25 for the SP02. However, differences were observed in $\tau(0.500 \mu m)$ during the summer months. The CIMEL (Fig.2a) showed higher mean values with greater variability compared to the SP02(Fig.2b). These differences can be attributed to the distinct time periods covered by the two instruments (see Table 1) and the occurrence of extreme wildfire events in North America and Siberia between 2019 and 2020 (Pulimeno et al. (2024), Engelmann et al. (2021), Ohneiser et al. (2021)), which were only captured by AERONET with the CIMEL.

Further evidence for the impact of wildfire smoke on the arctic aerosol composition was found at the Toolik Lake station by Welch et al. (2025).



At Oliktok, measurements spanned from September 2013 to June 2021, with lunar data measurements started in September 2018. Observations from Toolik Lake, taken from February 2017 to December 2024, do not include lunar data (see Table 1). Spectral $\tau(0.50\mu m)$ and $\alpha$ measurements for these stations are shown in Figure 3. Similar results from Barrow were observed
at Oliktok station, where the monthly mean values of $\tau(0.50\mu m)$ increase from 0.05 in February to around 0.09 in July, then decrease until reaching 0.04 in November, with $\sigma$ exceeding 0.08 in March, July, and August, and below 0.05 in other months. For the $\alpha$, Barrow and Oliktok exhibit similar behavior; the monthly mean values of $\alpha$ at Oliktok varying from 0.90 in February to 1.60 in July, with $\sigma$ equal to 0.35 on average.

At Toolik Lake station the monthly mean values of $\tau(0.50\mu m)$ exhibit a similar pattern to those observed at the other two
stations, ranging from approximately 0.04 in February to 0.14 in July. The standard deviation $\sigma$ was greater than 0.40 in July and less than 0.04 in the other months. In contrast, the $\alpha$ values were relatively stable throughout the seasons with minor oscillations, ranging from 1.28 in winter to 1.66 in summer, with $\sigma$ consistently below 0.40. Figure 3 also presents the Relative Frequency Histograms (RFH) of the daily mean values of $\tau(0.500\mu m)$ and $\alpha$ measured across all seasons. The RFHs for these stations were similar, although there were noticeable discrepancies between the means and percentiles. During winter,
the seasonal mean values of $\tau(0.500\mu m)$ were 0.07 for Barrow and Oliktok, and 0.04 for Toolik Lake. In contrast, during the summer season, the mean values were 0.18, 0.13, and 0.18 for the same stations. Consistent with the findings of Tomasi et al. (2015), the RFHs exhibited long tails towards higher $\tau$ values in all seasons, with particularly pronounced tails in winter and summer. This feature observed during winter may be attributed to higher AOD values in March and April, a consequence of the Arctic Haze phenomenon. The summer long-tail feature can be explained by the positive trend in total AOD during June, July,
and August (JJA), quantified as +0.007 AOD per decade by Xian et al. (2022). Their study demonstrated that during JJA, the smoke AOD contribution to the total AOD turns positive, with a +22% contribution per decade. This trend is visually evident in the multi-year monthly boxplots for June, July, and August shown in Figure 2 and 3 for all the North-American stations.





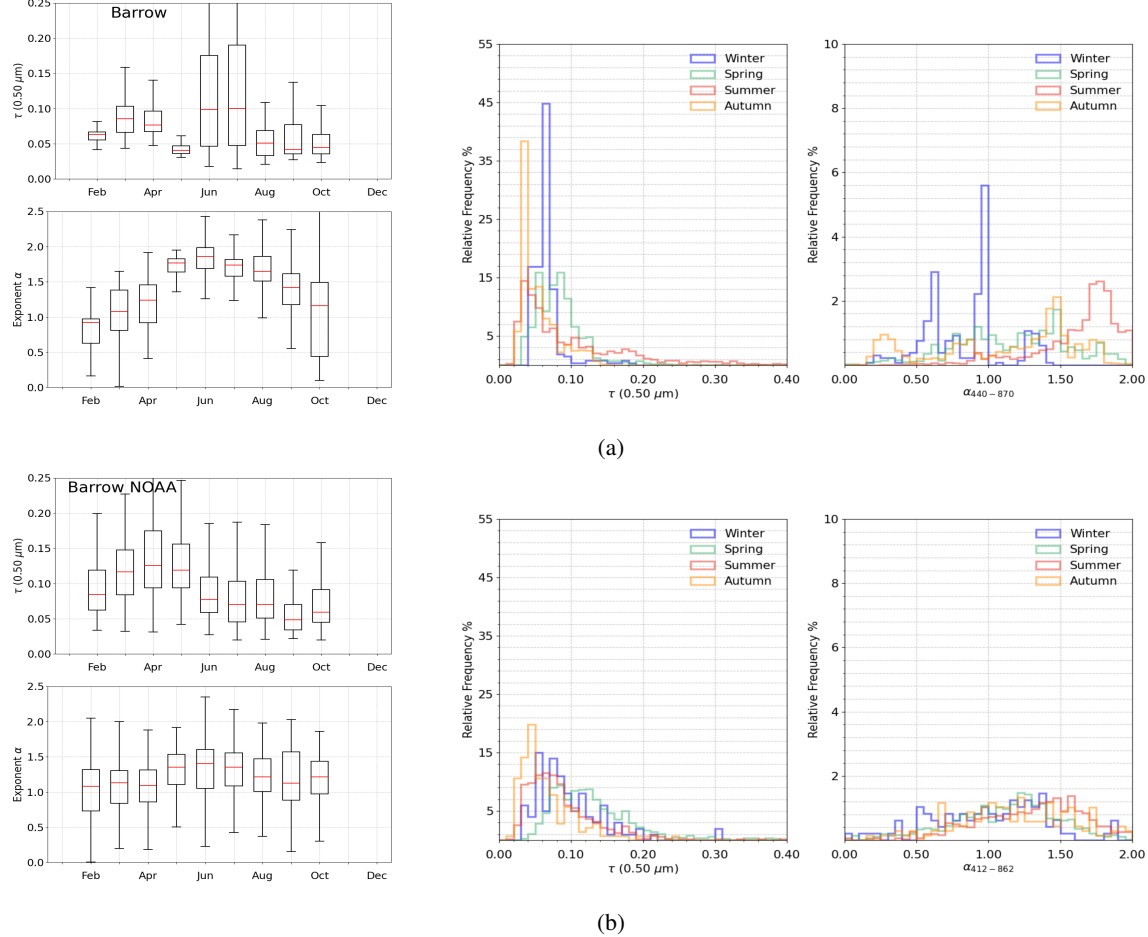

**Figure 2.** Left-hand side: Time-patterns of the monthly mean values and the first and third quartiles of aerosol optical thickness $\tau(0.500\mu m)$ and the Ångström exponent $\alpha$. The whiskers extend from the box to the farthest data point within 1.5 times the interquartile range (IQR) from the box. Right-hand side: relative frequency histograms for $\tau(0.500\mu m)$ and $\alpha$ separately for winter (arctic haze, DJF), spring (MAM), summer (background, JJA), and autumn (SON). These statistics were obtained from multi-year sun-photometer measurements conducted at Barrow by (a) Atmospheric Radiation Measurements, and by (b) National Oceanic and Atmospheric Administration.





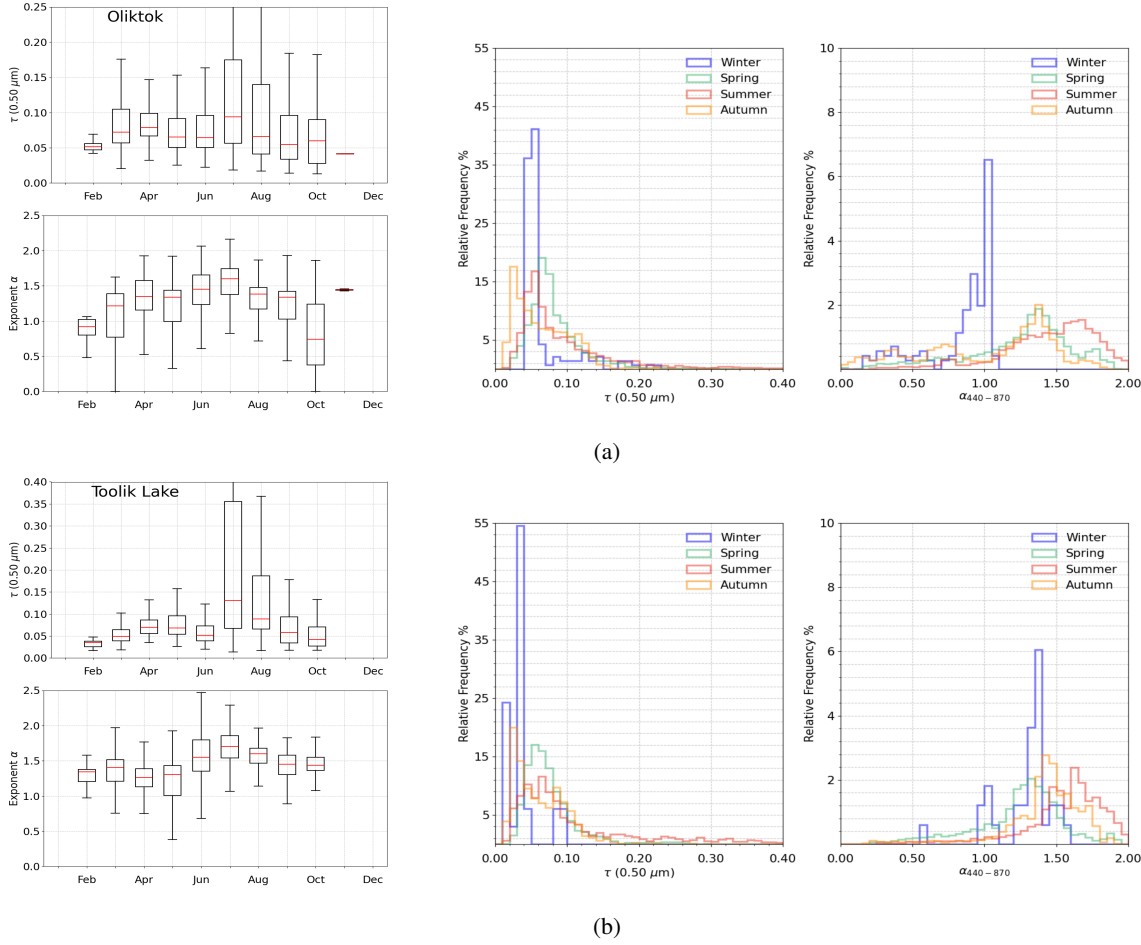

**Figure 3.** As in Fig.2, for the multi-year sun-photometer measurements of aerosol optical depth $\tau(0.500\mu m)$ and exponent $\alpha$ conducted at: (a) Oliktok by Atmospheric Radiation Measurements, and (b) Toolik Lake by National Ecological Observatory Network. To show the monthly boxplot of $\tau(0.50\mu m)$, the y-axis scale for the Toolik Lake station has been adjusted to range from 0 to 0.40.



## 3.2 Canada

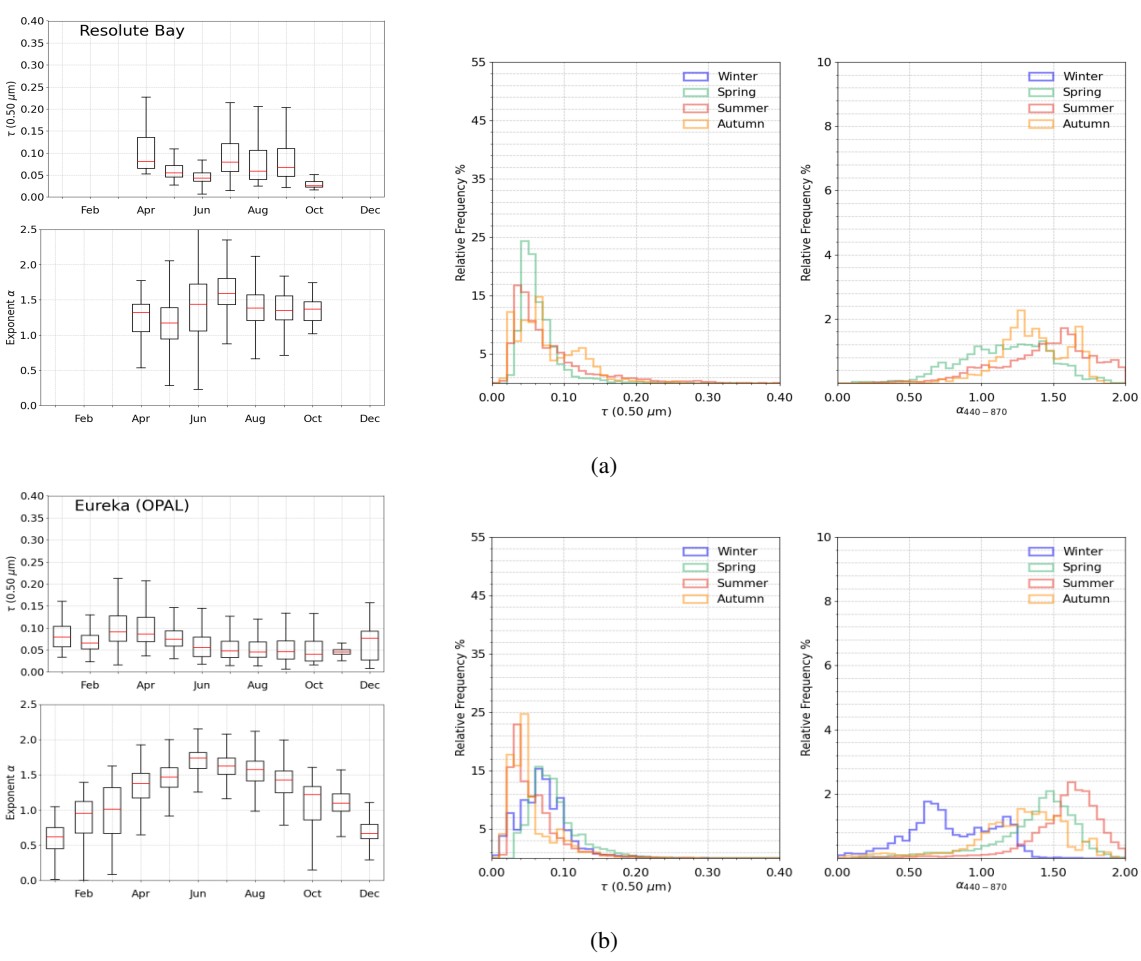

**Figure 4.** As in Fig.2, for the multi-year sun-photometer measurements of aerosol optical depth $\tau(0.500\mu m)$ and exponent $\alpha$ conducted at: (a) Resolute Bay by AEROCAN; (b) Eureka (OPAL).

The results from AERONET measurements conducted in the Canadian Arctic at Resolute Bay (July 2004 to December 2023)
and Eureka (OPAL) (April 2007 to December 2023) are shown in Fig.4. At Resolute Bay, a CIMEL CE318 sun-sky photometer measured aerosol optical thickness during the spring-summer period, when the solar airmass was above the acceptance threshold. At Eureka, a sun-sky-lunar photometer (model CE318-T) was installed in April 2016, allowing for additional measurements of $\tau$ and exponent $\alpha$ during the polar night.

At Resolute Bay, the monthly mean values of $\tau(0.500\mu m)$ are higher and more variable during the spring, ranging between
0.06 and 0.08, and show a decreasing trend in the following months. This decline should continue until reaching a minimum mean value of approximately 0.03 in October, as observed by Tomasi et al. (2015). Since air masses at this site primarily carry



aerosol particles from the North American continent (Hirdman et al. (2010)), the relatively elevated monthly mean $\tau$ values observed in July, August, and September may be attributed to an increased frequency of wildfire events affecting forested regions of Canada during the summer months (Jain et al. (2024)).

The Eureka (OPAL) station, being more remote, seems to be less impacted by local sources compared to Resolute Bay. This is reflected in the monthly mean values of the exponent $\alpha$. During the summer, $\alpha$ values at Eureka range between 1.30 and 1.50, indicating an atmosphere dominated by the presence of smaller particles. The values steadily decrease to around 0.50 in the other months, reaching their lowest during December and January, when the Arctic haze is most intense and dominated by larger particles. There may also be increased cirrus cloud contamination in the polar night data during winter at the Eureka site,

due to the less robust cloud-screening procedures that can be applied in the absence of aureole data at night. At Resolute Bay, the monthly mean values of $\alpha$ show smaller variations, staying around 1.40, suggesting the dominance of a relatively uniform background aerosol population, rather than indicating a complex mixture of aerosols; the lack of moon-photometer data during the polar winter forestalls any characterization of $\alpha$ variations. Fig.4 also indicates that the RFHs of $\alpha$ are asymmetrical. At Eureka, the winter seasons exhibit a bimodal distribution of values, with peaks at 0.60 and 1.15. In contrast, the other seasons

display wider left-hand tails, with higher values during summer, ranging from 1.48 (Resolute Bay) to 1.60 (Eureka).

In the Canadian Arctic, two additional stations, Alert and Cambridge Bay, were studied but not included in the main body of this paper. Both stations are managed by Environment and Climate Change Canada - AEROCAN, and use the sun-sky-lunar CIMEL CE318-T instrument to acquire aerosol extinction properties data. However, $\tau(0.500\mu m)$ and $\alpha$ are not presented in this paper for two main reasons: (i) the solar data at both stations were only available at level 1.5, whereas level 2 AERONET

data were used for the other stations in this study. (ii) There were insufficient measurements available, particularly between March and September 2024 at Alert, and between September and October 2024 at Cambridge Bay. As a result, it was not possible to retrieve statistically representative multi-annual monthly mean values for these parameters at these stations.

### 3.3   Greenland

Fig.5 shows the results from multi-year sun-photometer measurements conducted at three locations in Greenland: (i) Thule in

central-west coast of Greenland, (ii) Kangerlussuaq on the southwest coast, and (iii) Ittoqqortoormiit on the east coast. All sites are managed by NASA Goddard Space Flight Center (see Table 1 for more details).

At Thule, the monthly mean $\tau(0.500\mu m)$ values slowly decreased from around 0.07 in March-April-May to about 0.03 in October-November. The exponent $\alpha$ remained stable between March and September, with values between 1.30 and 1.60, before dropping sharply to around 0.60 in November-December due to the arrival of polluted air masses. The winter RFH for

$\tau(0.500\mu m)$ had a mean value of 0.04, with an almost-symmetrical shape and a long right-hand tail, influenced by Arctic haze episodes. The RFH at Thule for $\alpha$ shows an almost symmetrical distribution of data during spring, summer, and autumn. During winter, however, the measurements were concentrated within a smaller range of values. The mean $\alpha$ dropped from between 1.32 and 1.55 in spring and summer to 0.66 in winter. This suggests that there were no significant changes in aerosol composition, except in winter. However, the winter $\alpha$ value appears too low for pollution aerosols, which are typically dominated by fine-





mode particles. Therefore, it is more likely that the reduction in $\alpha$ is due to contamination from cirrus clouds in the nighttime data.

    The same pattern for $\alpha$ can be observed at Ittoqqortoormiit, with an almost symmetrical distribution of $\alpha$ during spring and autumn, having mean values of 1.24 and 1.17, respectively. winter values seem less influenced by coarse particles, with a mean of 0.73, which is slightly higher than at Thule.

On the other hand, the Kangerlussuaq station appeared less impacted by Arctic haze. The mean value of $\alpha$ during winter was 1.08, and it increased to 1.52 in the summer season. These features differ slightly from those reported by Tomasi et al. (2015), mainly due to the larger dataset analyzed in this study. More importantly, the current dataset benefits from the ability to close data gaps during the polar night, which was previously not possible.



**Figure 5.** As in Fig.2, for the multi-year sun-photometer measurements of aerosol optical depth $\tau(0.500\mu m)$ and exponent $\alpha$ conducted at: (a) Thule, (b) Kangerlussuaq and (c) Ittoqqortoormiit by NASA Goddard Space Flight Center. To show clearly the RFH of $\tau(0.50\mu m)$, the y-axis scale for the Ittoqqortoormiit station has been adjusted to range from 0 to 75.



### 3.4 Svalbard Archipelago

The Ny-Ålesund station can be described as a 'super-station' due to the collaboration of several institutions conducting pho-tometric studies using various types of photometers. Measurements taken at Ny-Ålesund by the AWI and the University of Valladolid using a CIMEL CE318-T, by the AWI with an SP1A, and by PMOD with a PFR are presented in Fig.6.

In 2017, the AWI, in partnership with the University of Valladolid, installed a sun-sky-lunar photometer. Data were collected from June 2017 to December 2023 (excluding December), providing valuable insights even during the polar night. The monthly

mean values of $\tau(0.500\mu m)$ at Ny-Ålesund ranged from 0.06 to 0.07 during March and April, gradually decreasing to about 0.04 in June and October. The RFHs of $\tau(0.500\mu m)$ exhibited a typical asymmetrical shape with long right-hand tails. The distribution was narrower during the winter and became wider in the spring and summer seasons. The monthly mean values of $\alpha$ increased from around 1.30 in March to 1.70, remaining relatively stable until September. However, following the polar dome's expansion toward mid-latitudes and the intrusion of polluted air masses, $\alpha$ values dropped from 1.00 in October to

around 0.90 in February. In line with the findings of Tomasi et al. (2015), the RFHs of $\alpha$ showed more dispersed features during winter, with a bimodal distribution shifted towards smaller values. This reflects greater variability in the aerosol mix during Arctic haze events, which bring fine particles from long-range transport and coarse particles from local sources or sea spray. During summer and autumn, $\alpha$ values ranged between 1.40 and 1.70, largely due to the increased variability of fine-mode particles, primarily driven by wildfire emissions with respect to coarse mode particles.

AOD was routinely measured at Ny-Ålesund also by AWI between 2012 and 2023 using an SP1A sun photometer (Fig. 6b). The measurements covered the period from March to October and were taken at 10 wavelengths ranging from 0.369 to 1.023 $\mu m$ with a high temporal resolution of 1 minute. AWI reported that the instrument was regularly calibrated at Izaña (Spain) using the Langley method. To make the observations comparable to those from AERONET, hourly averaged data for $\tau(0.500\mu m)$ were used. The $\alpha$ was calculated using Equation 4 in the spectral range of 0.412–0.860 $\mu m$. The AWI monthly

mean values of $\tau(0.500\mu m)$ ranged from 0.06 to 0.07 in spring and decreased to 0.04 during summer. A comparison of the AWI and AERONET results showed good agreement, despite differences in time periods (2012–2023 for AWI and 2017–2023 for AERONET) and coverage (AERONET also measured aerosol extinction properties during nighttime). The AWI monthly mean values of $\alpha$ were similar to those measured by AERONET, increasing from 1.30 in March to 1.50 in July and then gradually decreasing to 1.30 in September. Comparable seasonal mean values of $\tau(0.500\mu m)$ and $\alpha$ were observed between

AWI and AERONET, with a maximum difference of 0.01 in spring for $\tau(0.500\mu m)$ and 0.06 in summer for $\alpha$.

The PMOD measured aerosol extinction properties at Ny-Ålesund using a PFR from May 2002 to September 2023 (Fig. 6c), with measurements recorded every minute using four narrow-band interference filters centered at 0.368, 0.412, 0.500, and 0.862 $\mu m$. The $\alpha$ exponent was calculated within the spectral range of 0.412–0.862 $\mu m$. The parameters were averaged hourly. The PFR was calibrated annually at PMOD/WRC in Davos, Switzerland, during the polar winter. The monthly mean

values of $\tau(0.50\mu)$m and their seasonal variations align with results from other instruments. Specifically, $\tau(0.50\mu)$m averaged around 0.08 between March and May, coinciding with the final episodes of Arctic Haze. During the summer, the monthly mean decreased to a minimum of about 0.04 in September, when the Arctic atmosphere is typically cleaner.







**Figure 6.** As in Fig.2, for the multi-year sun-photometer measurements of aerosol optical depth $\tau(0.500\mu m)$ and exponent $\alpha$ conducted at Ny-Ålesund: (a) by AWI and University of Valladolid using a CIMEL CE318-T, (b) by AWI using a SP1A, and (c) by PMOD and Norwegian Polar Institute using a PFR.





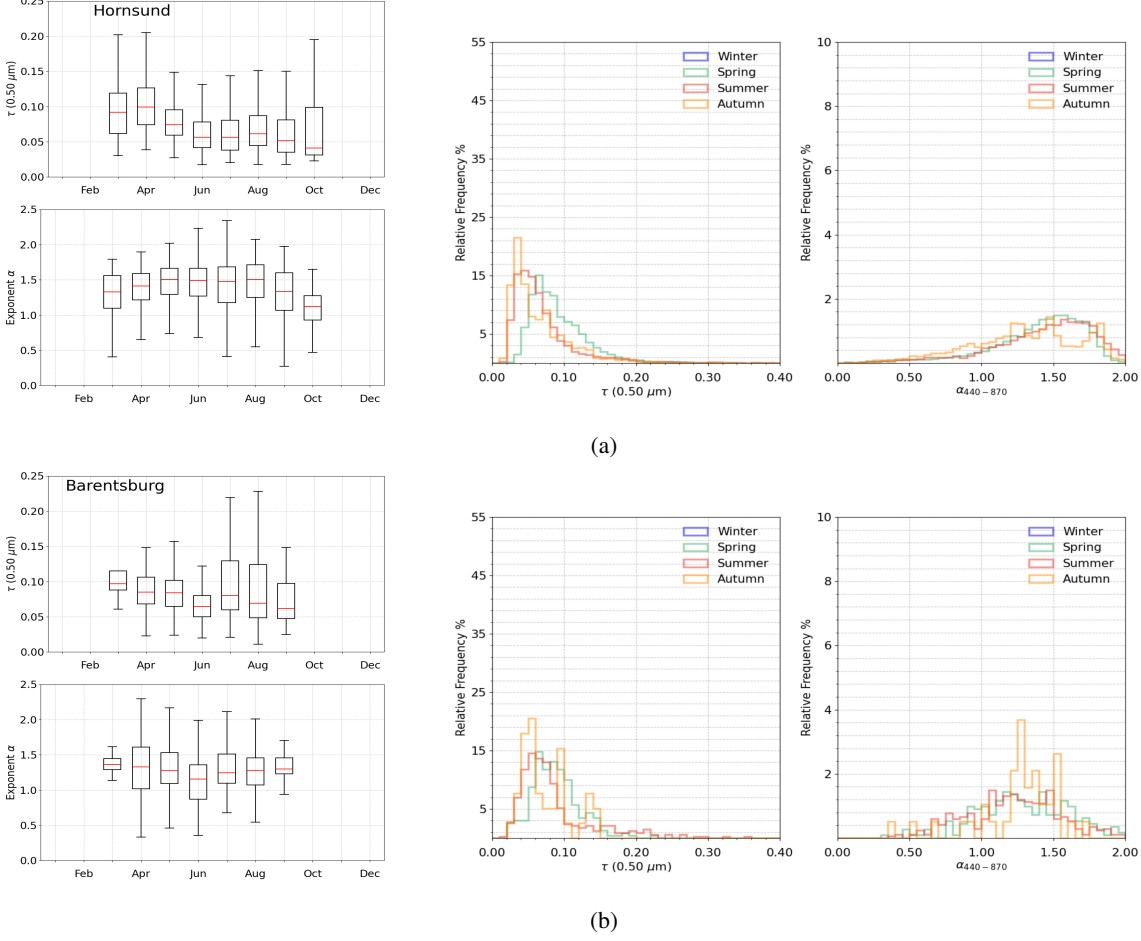

**Figure 7.** As in Fig.2, for the multi-year sun-photometer measurements of aerosol optical depth $\tau(0.500\mu m)$ and exponent $\alpha$ conducted at: (a) Hornsund by Polish Academy of Sciences and (b) Barentsburg by Russian Academy of Sciences.

A subsequent increase was observed in October and towards the winter season. Unfortunately, the PFR only provided sunphotometer data. The monthly mean values of $\alpha$ were also highly consistent, ranging between 1.30 and 1.50 from March to

October. The RFHs of $\tau(0.50\mu)$m showed an asymmetrical distribution with a right-skewed tail, with a mean value of 0.07 in summer and 0.09 in spring. Similarly, the RFHs of $\alpha$ displayed an asymmetrical distribution, with a left-skewed tail toward smaller values, and had a consistent shape and mean value of about 1.40 across all seasons.

The results observed at Ny-Ålesund closely matched those observed at Hornsund, even though the two sites are 200 kilometers apart. The Hornsund station collected a longer data series, covering the period from May 2004 to December 2023, with

measurements taken between March and October. As shown in Fig.7a, Hornsund displayed maximum monthly mean values of $\tau(0.500\mu m)$, ranging from 0.09 in March to 0.10 in April, followed by a decline through the summer and autumn, reaching about 0.04 in October. The monthly mean values of $\alpha$ at Hornsund remained stable between March and August, ranging from



1.30 to 1.50, with only small differences compared to Ny-Ålesund. The Hornsund RFHs of $\tau(0.500\mu m)$ show that the spring mean values were higher than those obtained during summer and autumn by more than 0.03. Even in this case, the shape of

the distributions were in good agreement with those reported at Ny-Ålesund. Unfortunately, no data are available for this site during winter.

At Barentsburg, the Arctic and Antarctic Institute, jointly with the Institute of Atmospheric Optics, carried out daily data collection from 2011 to 2023 (see Table 1) using a solar portable photometer SPM and its analogue with a sun tracker system (SP-9). These devices were developed to monitor the spectral transparency of air in the range 0.340–2.140 $\mu m$. In line with

other stations, aerosol optical depth was investigated at 0.500 $\mu m$, while the Ångström exponent was calculated in the spectral range 0.440–0.870 $\mu m$. The measurements are shown in Fig.7b. The monthly mean values of $\tau(0.500\mu m)$ decreased from approximately 0.10 in March to 0.07 in September, reflecting the typical behavior of an Arctic site. During the same period, $\alpha$ varied between 1.20 and 1.40. The RFHs of daily mean $\tau(0.500\mu m)$ showed similar mean values across all seasons, with 0.09 in spring and summer and 0.07 in autumn. However, these RFHs exhibited elongated right tails, indicating occasionally high

$\tau(0.500\mu m)$ values. In contrast, the RFHs of $\alpha$ followed a Gaussian distribution, with mean values smaller than those reported by other stations in the Svalbard Archipelago. Specifically, the mean $\alpha$ was 1.32 during spring, 1.23 during summer, and 1.28 during autumn. The spread of $\tau(0.500\mu m)$ values observed during July and August highlights the significant influence of smoke aerosols from boreal wildfires, as previously reported by Kabanov et al. (2023). The phenomenon of summer forest fires was most pronounced during the period 2010–2020. As a result, it is more evident at Barentsburg than at the other stations

in Svalbard, although it can still be observed elsewhere.

### 3.5 Scandinavia

Fig.8 presents the time patterns of monthly mean values for $\tau(0.500\mu m)$ and $\alpha$, as well as the seasonal RFHs for both parameters, derived from the Scandinavian stations at Andenes, Sodankylä, and Matorova (see Table 1 for details). The behavior of these parameters differed from those observed at other Arctic sites, as shown in Figs. 2 to 7.

At Andenes, the monthly mean values of $\tau(0.500\mu m)$ increased from 0.03 in February to 0.06 in April, remaining relatively stable during the summer and autumn before slowly decreasing to 0.04 by October. The seasonal variation from winter-spring to summer-autumn was less pronounced. The monthly mean values of $\alpha$, consistent with the findings of Tomasi et al. (2015), increased from 0.70 in January to 1.60 in July, followed by a decline until December.

At Sodankylä, the monthly mean values of $\tau(0.500\mu m)$ stayed relatively stable between February and September, around

0.06, before decreasing to about 0.04 in October-November and rising again to 0.08 in December-January. The trend for $\alpha$ exhibited a bell-shaped pattern, similar to Andenes, increasing from 0.90 in February to 1.80 in July. At Matorova, data were available for the whole year, covering the period from September 2020 to September 2024. The monthly mean values trend of $\tau(0.500\mu m)$ showed a bell shape, with minimum values around 0.03 in February and November, and maximum values of about 0.07 during late spring and summer. The $\alpha$ trend followed a similar pattern to the other Scandinavian stations, peaking

at 1.70 in summer and reaching a minimum of around 0.80 in January, in the middle of the winter season. Despite the different measurement periods at these stations—February 2002 to December 2023 for Andenes, March 2013 to December 2023 for



Sodankylä, and September 2020 to September 2024 for Matorova—the RFHs for both $\tau(0.500\mu m)$ and $\alpha$ displayed similar characteristics. The RFHs for $\tau(0.500\mu m)$ exhibited an asymmetrical shape with a wider distribution compared to those of other Arctic stations. Meanwhile, the RFHs for $\alpha$ showed a broader distribution of values across all seasons, with winter values displaying a bimodal shape and a distribution over smaller values. Due to the proximity of these Arctic stations to the Euro-Asian continent and the alternation of polluted air masses with sea-salt particles from ocean regions, the monthly mean values of $\tau(0.500\mu m)$ remained stable throughout the year, without exhibiting the typical seasonal trend expected for Arctic stations. When comparing Andenes and Sodankylä, it is found that the overall AOD values are generally similar, mainly because both sites are affected by similar regional air mass transport, despite their different local aerosol types. Sodankylä, as a clean continental site, typically shows higher $\alpha$ and lower coarse mode concentrations, while Andenes, being a coastal site, is influenced by marine aerosols and sea salt, which maintain a higher coarse-mode contribution throughout the year (Rodríguez et al. (2012); Toledano et al. (2012)).

In the Russian Federation, only the Cape Baranova station has been studied but not shown, consistent with the approach taken for the Arctic sites of Alert and Cambridge Bay. At Cape Baranova, the Russian Academy of Sciences investigated aerosol extinction properties using an SP-9 instrument, the same model employed at Barentsburg. The measurements were recorded from April 2018 to August 2021, with a total of 59 daily observations. The mean values of $\tau(0.500\mu m)$ and $\alpha$ were 0.081±0.045 and 1.67±0.34, respectively. Due to the limited number of measurements and the lack of data for 2020, it was not possible to assess the aerosol climatology at this station.





**Figure 8.** As in Fig.2, for the multi-year sun-photometer measurements of aerosol optical depth $\tau(0.500\mu m)$ and exponent $\alpha$ conducted at: (a) Andenes by Andøya and University of Valladolid, (b) Sodankylä and (c) Matorova by the Finnish Meteorological Institute.





## 4 Measurements in Antarctica

Table 2 presents information about the eleven Antarctic sites discussed in this paper. The geographical locations of these stations are shown in Fig. 1b. As for the Arctic, we use level 2 data for solar photometry and level 1.5 data for lunar photometry, both from AERONET. For the Antarctic sites, where the seasons are reversed, the austral summer was defined as December to February; autumn (March to May); winter (June to August); spring(September to November).

The main difference between the Arctic and Antarctica is the distribution of stations across the poles. Since Antarctica is
more isolated from other continents, there are very few stations for photometry observations. Most of the sites are located on the Antarctic Peninsula because it's closer to South America, allowing for interesting measurements due to favorable atmospheric conditions. Six other stations are in the Atlantic sector of Antarctica, and two are located on the Antarctic Plateau (Fig. 1b). Due to the scattered and uneven distribution of sites in Antarctica, it is difficult to get a complete picture of aerosol evolution and behavior on the continent.

### 4.1 Antarctic Peninsula

The results from sun-photometer measurements taken at the Antarctic Peninsula, specifically at Escudero, Juan Carlos I, and Marambio stations, are shown in Fig.9 and 10.

| Stations - ID | Managing Institutions | Instrument model | Coordinates and Altitude | Measurement Period | Lunar Data started |
|---|---|---|---|---|---|
| Antarctic sites | | | | | |
| Dome C - 10 | National Research Council, Italy - NOAA, US | SP02 and CMDL | 75.1800 S, 123.3800 E, 3233 m a.m.s.l. | November 2010 - February 2020 [3,735] | no |
| Escudero - 2 | University of Santiago, Chile - Chilean Antarctic Institute, Chile | CE318 | 62.2015 S, 58.9657 W, 33 m a.m.s.l. | December 2019 - December 2023 [2,616] | February 2020 [356] |
| Juan Carlos I - 1 | University of Valladolid, Spain | CE318 | 62.6630 S, 60.3894 W, 5 m a.m.s.l. | December 2022 - December 2023 [487] | March 2023 [25] |
| Marambio - 3 | University of Valladolid, Spain | CE318 | 64.2400 S, 56.6252 W, 200 m a.m.s.l. | February 2008 - December 2023 [15,204] | January 2018 [2,344] |

*Continued on next page*



| | | Antarctic sites (continued) | | | |
|---|---|---|---|---|---|
| Stations - ID | Managing Institutions | Instrument model | Coordinates and Altitude | Measurement Period | Lunar Data started |
| Marambio - 3 | Finnish Meteorological Institute, Finland - Physical Meteorological Observatory Davos, Switzerland | PFR | 64.2400 S, 56.6252 W, 200 m a.m.s.l. | February 2005 - May 2016 [2,748] | no |
| Mirny - 9 | Arctic and Antarctic Research Institute - Zuev Institute of Atmospheric Optics, Russia | SPM | 66.3300 S , 93.0100 E, 40 m a.m.s.l. | January 2013 - May 2023 [939] | no |
| Neumayer 3 - 4 | Leibniz Institute for Tropospheric Research, Germany - Alfred Wegener Institute, Germany | CE318 | 70.6652 S, 8.2836 W, 65 m a.m.s.l. | January 2023 - December 2023 [5,494] | March 2023 [818] |
| Syowa - 7 | Japan Meteorological Agency, Japan | MS110 and PFR | 69.0053 S, 39.5811 E, 29 m a.m.s.l. | January 1996 - December 2020 [6,098] | no |
| South Pole - 11 | NASA Goddard Space Flight Center, US | CE318 | 90.0000 S, 70.3000 E, 2835 m a.m.s.l. | November 2007 - December 2023 [19,595] | no |
| South Pole - 11 | NOAA, US | SP02 | 90.0000 S, 70.3000 E, 2835 m a.m.s.l. | November 2001 - March 2014 [1,894] | no |
| Troll - 5 | NILU, Norway - Physical Meteorological Observatory Davos, Switzerland | PFR | 72.0167 S, 2.5333 E, 1309 m a.m.s.l. | January 2007 - December 2023 [898,339] | no |
| Trollhaugen - 5 | NILU, Norway - Physical Meteorological Observatory Davos, Switzerland | PFR | 72.0117 S, 2.5351 E, 1553 m a.m.s.l. | January 2007 - December 2023 [898,339] | no |

*Continued on next page*





| Stations - ID | Managing Institutions | Instrument model | Coordinates and Altitude | Measurement Period | Lunar Data started |
|---|---|---|---|---|---|
| | | Antarctic sites (continued) | | | |
| Utsteinen - 6 | Royal Belgian Institute for Space Aeronomy, Belgium | CE318 | 71.9500 S, 23.3333 E, 1396 m a.m.s.l. | February 2009 - December 2023 [12,992] | no |
| Vechernaya Hill - 8 | National Academy of Sciences of Belarus, Belarus - University of Lille, France | CE318 | 67.6600 S, 46.1580 E, 80 m a.m.s.l. | December 2008 - December 2023 [7,846] | no |

Table 2: List of Antarctic stations using different models of photometer. For each station, the coordinates and altitude are specified, along with the measurement period for solar photometry and the installation period of the lunar model. The number of measurements for both the solar and lunar periods is provided in parentheses.

At Escudero station, data was collected from December 2019 to December 2023, showing greater variability in monthly
mean values of $\tau(0.500\mu m)$. The AOD ranges from around 0.04 in December to 0.11 in September. The RFHs are similar to those at Marambio, with an asymmetrical right-hand shape for $\tau(0.500\mu m)$ and a more symmetrical distribution for $\alpha$. These findings are consistent with those reported by Tomasi et al. (2015), where the authors attributed this pattern to the presence of sea salt particles, which dominate the extinction process.

Juan Carlos I station provided data from December 2022 to December 2023, representing the shortest timeseries of this
sector. Since this station operates only during the austral summer, measurements were taken only from December to March. The monthly mean values of $\tau(0.500\mu m)$ ranged from 0.06 in January to 0.09 in March. There was a decreasing trend in the first two months, followed by an increase in the last two months of the series. The monthly mean values of $\alpha$ were very low, ranging from about 0.30 in March to 0.60 in February, likely due to the particles being of sea salt origin. In general, since data is only available for the summer months, it is difficult to draw consistent conclusions about aerosol behavior. Due to the limited
number of measurements, the RFH of $\tau(0.500\mu m)$ shows an asymmetrical distribution in summer and unclear dispersion in autumn.. The mean value during summer was 0.08, with the 25th and 75th percentiles at 0.05 and 0.08, respectively. The RFH of $\alpha$ was also quite symmetrical in the 0.10–1.00 range, with a mean value of 0.48, and the 25th and 75th percentiles differed by less than 0.20 from each other.

The measurements at Marambio have been conducted by the University of Valladolid since February 2008 by using a
CIMEL sun-sky-lunar photometer, representing the longest time series in this sector of the continent. The monthly mean





values of $\tau(0.500\mu m)$ range from about 0.03 in October to 0.07 in July, while the exponent $\alpha$ varies from 0.60 in June to 1.40 in February. A seasonal trend is observed for $\alpha$, with higher values during the austral summer and lower values in winter.

The FMI also collaborates with Physical Meteorological Observatory Davos (PMOD) at the same station, measuring aerosol extinction parameters using a PFR sun-photometer. Quality assured aerosol optical depth and Ångström exponent for Marambio

GAW/PFR station, from February 2005 to May 2016, were provided by PMOD. Measurements were taken with a high temporal resolution of 1 minute at four wavelengths, namely 0.368, 0.412, 0.500, and 0.862 $\mu m$. The $\alpha$ was calculated for the spectral range of 0.412–0.862 $\mu m$. Processing and quality checks were performed according to the protocols of the World Optical Depth Research and Calibration Center (WORCC). The PFR sun-photometer collected data even during the winter-spring season, despite being a sun-based instrument. This was possible because of the station's location at 64°S, where some light is

still present during winter. The monthly mean values of $\tau(0.500\mu m)$ ranged from around 0.05 in August to about 0.03 in May, while $\alpha$ ranged from 0.75 in August to 1.50 in December. The AOD monthly time series at Marambio showed slight differences between the CIMEL and PFR instruments, although the overall patterns were similar, with maximum AOD values during late autumn and winter, and minimum values during summer. However, the monthly trends of $\alpha$ showed significant differences, particularly in the data distribution. These differences may be due to the use of different quality assurance methods applied

by AERONET for the CIMEL and by PMOD for the PFR. By analyzing the monthly mean values of $\alpha$ measured by both photometers at Marambio, as shown in Fig. 10, it is evident that coarse-mode particles likely dominate the atmosphere during the austral winter. In contrast, fine-mode particles are more prevalent in summer, possibly due to increased human activities and cruises in the peninsula. This seasonal shift is supported by previous studies reporting higher aerosol scattering coefficients in winter due to enhanced sea salt emissions and dominance of coarse-mode particles (Asmi et al. (2018)). A similar trend is

observed at the other two stations on the Antarctic Peninsula, though with slightly different magnitudes.





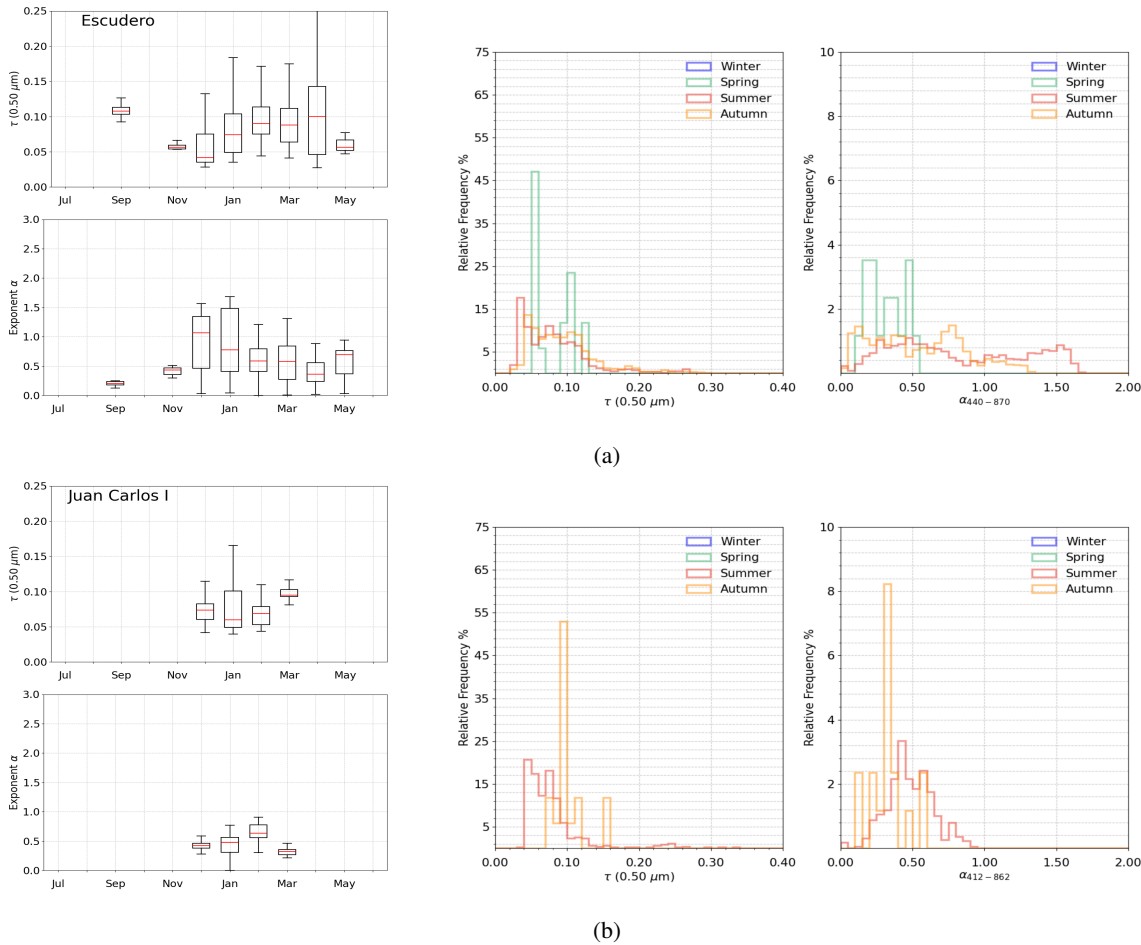

**Figure 9.** As in Fig.2, for the multi-year sun-photometer measurements of aerosol optical depth $\tau(0.500\mu m)$ and exponent $\alpha$ conducted at:
(a) Escudero by University of Santiago and Chilean Antarctic Institute, and (b) Juan Carlos I by University of Valladolid.





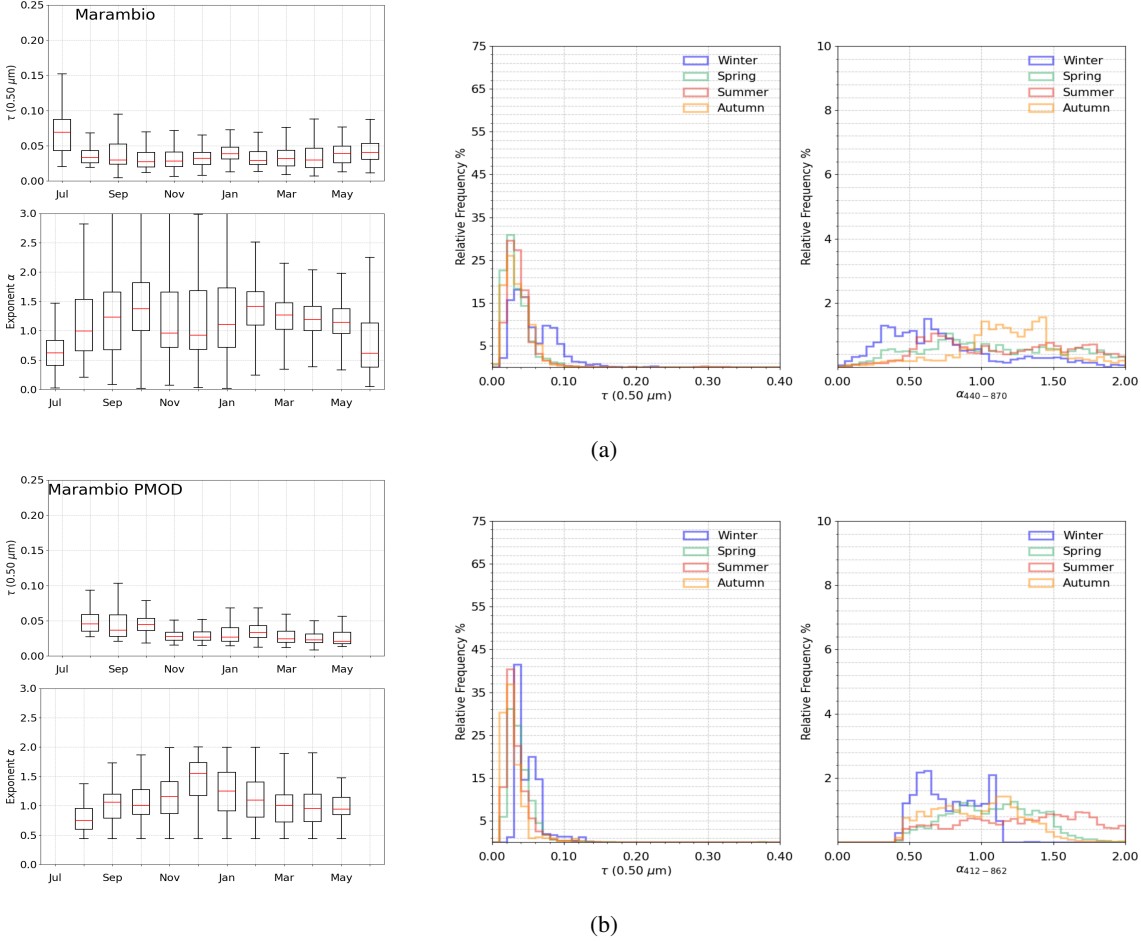

**Figure 10.** As in Fig.2, for the multi-year sun-photometer measurements of aerosol optical depth $\tau(0.500\mu m)$ and exponent $\alpha$ conducted at: (a) Marambio by the University of Valladolid using a CIMEL, and (b) Marambio by the Finnish Meteorological Institute and PMOD/WRC using a PFR.

## 4.2 Atlantic Sector

Fig. 11 and 12 show the results obtained from the measurements conducted at six other coastal sites in the Atlantic sector. The Neumayer measurements, conducted by the Leibniz Institute for Tropospheric Research in collaboration with the Alfred Wegener Institute, were conducted over the period November - April, showing rather stable time-patterns of the monthly mean values of $\tau(0.500\mu m)$, ranging from 0.03 to 0.05, and associated with stable values of $\alpha$ varying from 0.80 to 1.20. The RFH of $\tau(0.500\mu m)$ during summer showed a mean value of 0.04 and 25th and 75th percentiles of 0.03 and 0.04, respectively. The RFH for $\alpha$ appear symmetrical with little dispersion for all moths, with a mean value around 1.00 for all seasons.






The NILU, in collaboration with PMOD, operates a PFR sun-photometer at Troll/Trollhaugen, collecting data from January 2007 to December 2023. In February 2014, the instrument was relocated from Troll to the new Trollhaugen observatory due to local contamination. The monthly mean values of $\tau(0.500\mu m)$ show a steady trend throughout the year, with mean values ranging between 0.02 and 0.03. Similar to other stations in Queen Maud Land, located in the Atlantic sector of Antarctica, the monthly mean values of $\alpha$ were lower in winter and spring and higher in summer. In general, the Antarctic baseline atmosphere is a mixture of descending free-tropospheric and lower-stratospheric air. The particle size distribution is dominated by Aitken-mode particles in summer and accumulation-mode particles in winter (Fiebig et al. (2009); Fiebig et al. (2014)). These features are illustrated in Fig. 11 and Fig. 12. The RFHs of $\tau(0.500\mu m)$ show an asymmetrical distribution skewed toward smaller values across all seasons. For $\alpha$, the RFHs display a Gaussian distribution in spring and autumn, with mean values of 1.29 and 1.35, respectively. In winter, the distribution is asymmetrical with a right-hand skew and a mean of 0.96. In summer, the distribution is asymmetrical with a left-hand skew and a mean of 1.47.

The data from Utsteinen and Vechernaya Hill stations were even more limited, covering the periods from November to February and November to March, respectively (see Table 2 for additional info). In both cases, the monthly mean values of $\tau(0.500\mu m)$ remained very stable, around 0.02 at Utsteinen and 0.03 at Vechernaya Hill. The RFHs showed the typical pattern found at other Antarctic stations, with a symmetrical distribution leaning toward lower values and a right-hand skew. At Utsteinen, during summer season, the mean value of $\tau(0.500\mu m)$ was 0.02, with the 25th and 75th percentiles at 0.01 and 0.02, respectively. At Vechernaya Hill, the mean value was 0.03, with 25th and 75th percentiles at 0.02 and 0.03. The monthly boxplot of $\alpha$ at Utsteinen displayed quite stable values throughout the time series, and the related RFH had a symmetrical and even distribution across all values, indicating that the stable extinction features were mainly due to sea salt particles. Some differences were observed at Vechernaya Hill, where monthly mean values of $\alpha$ decreased from November to March. The monthly boxplot panel of $\alpha$ at this station showed high values, between 2.20 in November and 1.30 in March, showing that the aerosol extinction features are in part produced by fine mode particles (e.g. non-sea-salt sulphate aerosols).

The longest time series in the Atlantic sector was provided by the Japan Meteorological Agency at Syowa Station. The first Japanese permanent research base was established in 1957. Measurements were recorded using an MS110 sun-photometer from January 1996 to February 2011, and a PFR sun-photometer from February 2011 to December 2020. Although no lunar data were recorded at the station by either instrument, the winter period appeared to be present in the RFHs due to observations during the transition from night to day. To make the measurements comparable with those from AERONET, we averaged the data hourly and then calculated monthly statistics for atmospheric turbidity parameters. Specifically, we used $\tau$ measurements at 0.500 $\mu m$ and $\alpha$ in the spectral range of 0.412-0.862 $\mu m$. The monthly mean $\tau(0.500\mu m)$ values show no seasonality, remaining within the range of 0.02-0.03 during the period from August to April. The monthly boxplots for $\alpha$ clearly show the a predominance of fine aerosol particles in summer, with an average value of 1.43 during this season. In contrast, coarse particles, likely influenced by sea salt due to the station's location on an island 4 km from the coast, dominate in winter, with a seasonal mean value of 0.84.

Figure 12c presents the results of measurements conducted at Mirny using an SPM handheld sun photometer between January 2013 and May 2023, covering the period from August to May each year. This Russian device, developed by the



Institute of Atmospheric Optics, records incoming solar radiation at 10 different wavelengths ranging from 0.339 to 2.139 $\mu m$. To ensure comparability between stations, the 0.500 $\mu m$ wavelength was selected for AOD measurements, while the $\alpha$ was

calculated using the wavelength range of 0.443–0.871 $\mu m$. The monthly mean values of $\tau(0.500\mu m)$ ranged from 0.02 to 0.03, while the mean values of $\alpha$ varied from approximately 1.30 during the summer months to 0.80 in late winter and early spring. The RFHs of $\tau(0.500\mu m)$ showed nearly symmetrical peaks across all seasons, with mean values of 0.02 during all the season, with little differences. In contrast, the RFH of $\alpha$ exhibited a wider dispersion over the measured range. The seasonal mean and median values of $\alpha$ were consistently within the interval of 0.80–1.30. These findings indicate that aerosol extinction features

at Mirny were predominantly influenced by sea-salt particles, likely generated by wind activity over the Antarctic Ocean. This is a common feature observed at all Antarctic stations in the Atlantic region.




(a)

(b)

(c)

**Figure 11.** As in Fig.2, for the multi-year sun-photometer measurements of aerosol optical depth $\tau(0.500\mu m)$ and exponent $\alpha$ conducted at: (a) Neumayer by Leibniz Institute for Tropospheric Research and Alfred Wegener Institute, (b) Troll/Trollhaugen by NILU and Physical Meteorological Observatory Davos, and (c) Utsteinen by Royal Belgian Institute for Space Aeronomy.





**Figure 12.** As in Fig.2, for the multi-year sun-photometer measurements of aerosol optical depth $\tau(0.500\mu m)$ and exponent $\alpha$ conducted at: (a) Vechernaya Hill by National Academy of Sciences of Belarus and University of Lille (b) Syowa by the Japan Meteorological Agency, and (c) Mirny by the Arctic and Antarctic Research Institute.





### 4.3 Antarctic Plateau

The results obtained at Dome C from sun-photometer measurements conducted since 2010 are shown in Fig. 13a. From 2010 to 2012, the NOAA contributed data using a CMDL sun-photometer, a device capable of acquiring measurements at several spectral wavelengths: 0.368, 0.412, 0.500, 0.610, 0.675, 0.778, and 0.862 $\mu m$, with a reported accuracy of 0.02 (Kim et al. (2005)). Subsequently, from 2015 to 2020, the National Research Council of Italy and the NOAA carried out measurements of aerosol extinction properties at the same site using a Carter Scott SP02 sun-photometer. Since both instruments operated at the same wavelengths and at the same location, the data are presented together in this paper rather than separately, as done for other cases. The monthly mean values of $\tau(0.500\mu m)$ obtained from these sun-photometers remained stable throughout the investigation period, with a mean value of 0.02 in November, increasing to 0.04 in February. These low values, close to the detection limit of the instruments, can be attributed to the extremely clean air at Dome C, located on the Antarctic plateau at an altitude of over 3000 m, and to the predominant role of subsidence processes in influencing aerosol loads. The monthly mean values of the $\alpha$ during the same period ranged from approximately 1.50 to 1.80. The RFH of $\tau(0.500\mu m)$ exhibited an asymmetrical distribution, with a right-skewed tail toward higher values. The mean $\tau(0.500\mu m)$ was 0.03 during summer and spring. The RFH of $\alpha$ showed a dispersed distribution in the range of 0.00–2.00, with a mean of 1.60 during summer. The 25th and 75th percentiles of $\alpha$ were 1.35 and 1.91, respectively.

Fig.13b and Fig.13c show the measurements of $\tau(0.500\mu m)$ and $\alpha$ at Amundsen-Scott station, registered as South Pole on AERONET; the station is located at the geographical South Pole and managed by NOAA (see Table 2). Photometric measurements were taken by NASA at this site since November 2007 by using a CIMEL instrument, covering a good period of time, but data were available only during the Austral summer between November and February. Since this station is located in the center of the continent, far away from the coast, a cleaner atmosphere was aspected. This feature was confirmed by the monthly mean values of $\tau(0.500\mu m)$, showing steady behaviour, between 0.02 in November and 0.04 in February. The RFH of this parameter showed a symmetrical shape for both seasons available (spring and summer), with mean value of around 0.02 and 25th and 75th percentiles of 0.02 and 0.03 respectively. The monthly mean values of $\alpha$ showed a minimum of around 0.50 in November, followed by a relatively steady behavior in the subsequent months with values around 1.0, indicating the presence of fine-mode particles even at this remote site. It is important to note that there are very few data points for October and November at the South Pole NOAA station. Therefore, the data sample for these months is not robust, and the results may be affected by even small amounts of cloud contamination. The high number of personnel and scientific activities make this station one of the most active in the continent, and local pollution sources can explain these numbers, mainly aircraft flights and fuel usage for electricity generation (Sheridan (2015)). At the same site, NOAA monitored aerosol extinction features using a Carter Scott SP02 sun-photometer from November 2001 to March 2014. This instrument measured incoming solar radiation at four wavelengths: 0.412, 0.500, 0.675, and 0.862 $\mu m$. However, for this study, only the wavelength at 0.500 $\mu m$ was considered. The $\alpha$ was analyzed across the spectral range of 0.412–0.862 $\mu m$. Figure 13c illustrates the monthly mean values of $\tau(0.500\mu m)$, extending the coverage from September to March. The highest values occurred in early spring (September), with a mean of approximately 0.09, while the lowest values were observed during the austral summer (December), with a mean




of 0.02. The behavior of the $\alpha$ followed a similar trend, showing an increase from spring to summer, followed by a decrease towards autumn. Due to differences in the observation periods and the number of measurements available (see Table 2), the RFH of $\tau(0.500\mu m)$ displayed additional features compared to NASA's dataset. The distribution was asymmetrical, with a right-skewed tail towards higher values. The mean value of $\tau(0.500\mu m)$ was 0.07 during summer and 0.09 during spring.

Similarly, the RFH of $\alpha$ also showed differences. While NASA's data exhibited a bimodal distribution for both summer and spring, NOAA's measurements of $\alpha$ were randomly distributed across the interval, showing no dominant distribution pattern. It would be extremely interesting to understand how aerosol extinction properties evolve during the winter season at these Antarctic Plateau stations. However, due to the extremely low temperatures (as low as -70°C), designing a heating system that ensures the photometer's proper functionality is challenging.





**Figure 13.** As in Fig.2, for the multi-year sun-photometer measurements of aerosol optical depth $\tau(0.500\mu m)$ and exponent $\alpha$ conducted at: (a) Dome C by National Research Council of Italy and National Oceanic and Atmospheric Administration (NOAA), (b) South Pole by NASA/GSFC, and (c) South Pole by NOAA.





## 5   AOD trend analysis

Calculating trends is one of the most important and common tasks in climatological studies. The $\tau(0.500\mu m)$ trends have been calculated for several stations at both poles to gain a general understanding of how this parameter has changed over time. The analysis of $\alpha$ trends were not included in this study, because this parameter can be influenced by variations in both aerosol type and size distribution, making it challenging to derive robust conclusions from limited datasets, and because in polar regions $\tau$ values are very low leading to higher uncertainty in $\alpha$ retrievals.

In the Arctic, five stations were selected: Barrow representing North America; Eureka (OPAL) representing Canada; Thule for Greenland; Ny-Ålesund for the Svalbard Archipelago; and Andenes for the Scandinavian countries. In Antarctica, four stations were analyzed: Marambio representing the Antarctic Peninsula; Troll/Trollhaugen and Syowa for the Atlantic sector; and South Pole for the Antarctic Plateau.

For the trend analysis, two different approaches have been used: (i) the non-parametric Mann-Kendall test (Mann (1945), Kendall (1955)), in which the null hypothesis states that the data are identically distributed, while the alternative hypothesis suggests that the data follow a monotonic trend. (ii) the Theil-Sen method (Theil (1992), Sen (1968)), which, given a set of n $x, y$ pairs, calculates the slopes between all pairs of points. The Theil-Sen estimate of the overall slope is the median of these slopes. Both approaches provide confidence intervals even with non-normal data and are resistant to outliers. To compute the trend analysis, monthly mean values of $\tau(0.500\mu m)$ have been used. Deseasonalization techniques, such as Seasonal Trend decomposition using LOESS (STL), were not applied in this analysis due to the lack of data for several months of the year. Although the lunar photometry technique helps to fill historical $\tau$ gaps in polar datasets, it has been in use for less than a decade, remains sparsely distributed across the poles, and operates under optimal meteorological conditions for only about ten days per month.

Moreover, since the last work published by Tomasi et al. (2015), only three volcanic eruptions have occurred that could potentially affect the atmosphere at such high-latitude sites. These eruptions are: (i) the Raikoke volcanic island eruption in June 2019 (Russia), (ii) the Bezymianny stratovolcano eruption in May 2022 (Russia), and (iii) the Sheveluch volcano eruption in April 2023 (Russia). The Raikoke eruption is particularly relevant for the Arctic atmosphere, as it resulted in a mean stratospheric aerosol optical depth of 0.025 (Vernier et al. (2024), Sofieva et al. (2024)). In-depth studies on the other two eruptions in 2022 and 2023 have not been published yet. Since there is no information available on the tropospheric impacts of these events in the Arctic, making difficult for aerosol models to simulate the lifecycle of the particles, and since Antarctica did not appear to be influenced by the volcanic eruptions, the data has not been cleaned to account for these events.

Table 3 summarizes the key statistics from the trend analysis of all selected stations, as determined by the Mann-Kendall test. In this analysis, Kendall's tau serves as a non-parametric measure to evaluate the strength and direction of the trend, ranging from -1 (perfect negative correlation) to 1 (perfect positive correlation). The p-value indicates the significance level of the trend, with lower values signifying a trend that is statistically significant. Fig.14 shows the dominant trends identified by Theil-Sen regression for the Arctic station of Andenes, and the Antarctic stations of Syowa and South Pole. The trends for the other stations showed no significant results.





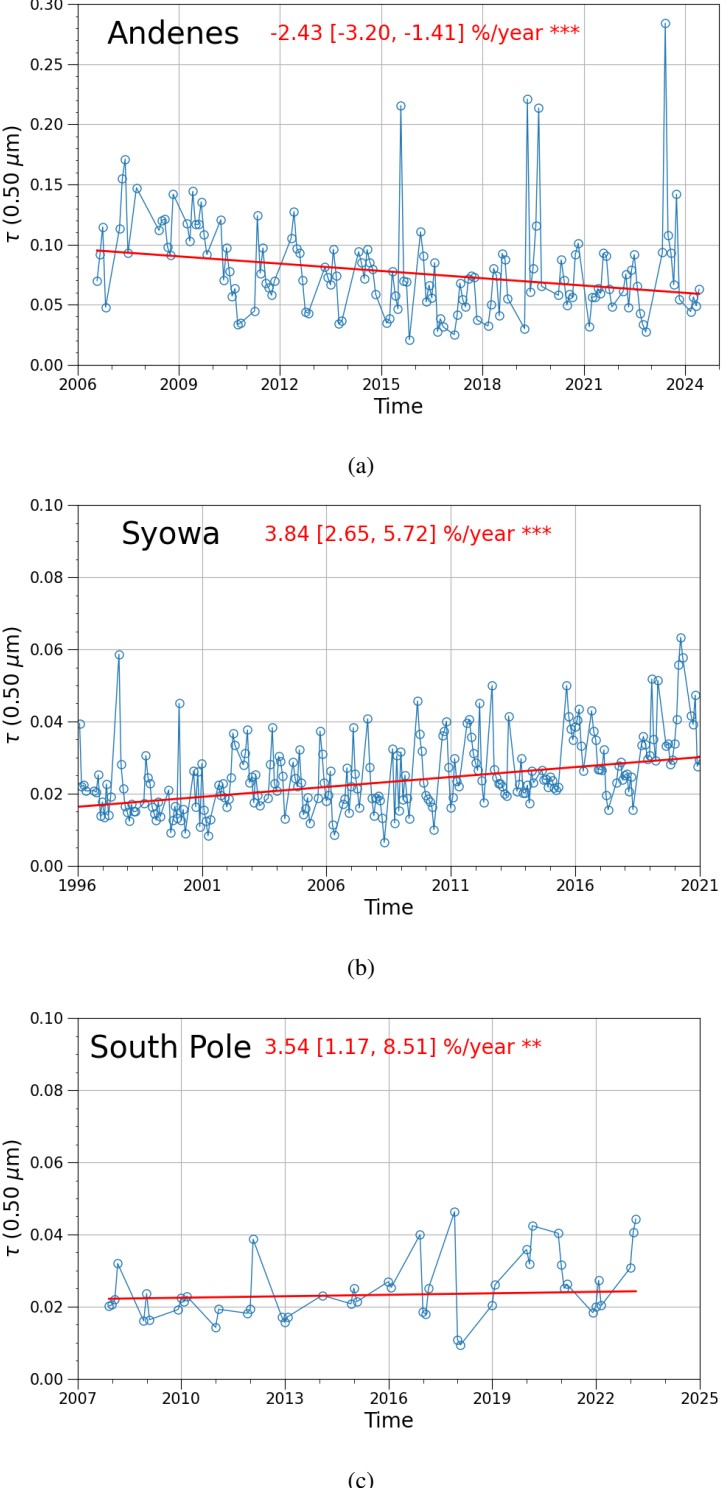

(a)

(b)

(c)

**Figure 14.** Theil-Sen slopes for mothly median $\tau(0.500\mu m)$ at (a) Andenes, (b) Syowa, and (c) South Pole. The solid red line shows the trend estimate; the overall trend in shown at the top as % per year, together with the 95% confidence intervals. The *** show that the trend is significant to the 0.001 level, the ** to the 0.01 level





Andenes shows a negative trend of -2.43% per year, with a Kendall's tau of -2.33e-01 and a p-value of 4.74e-05, indicating
a high level of statistical significance. This negative trend can be attributed to the strict regulations on industrial emissions
adopted by EU countries. Surprisingly, the Antarctic sites of Syowa and South Pole exhibit overall positive trends, with in-
creases of +3.84% per year and +3.54% per year, respectively. Syowa has a Kendall's tau of 0.327 and a highly statistically
significant of 3.02e-13, while South Pole has a Kendall's tau of 0.278 and a p-value of 5.40e-03. South Pole has relatively few
monthly measurements each year, as shown in Fig.14, and Syowa used an MS110 photometer from 1996 to February 2011,
which was replaced by a PFR instrument until 2020. Although fewer months are available for analysis in Antarctica, these
trends remain meaningful. Two possible explanations for these trends are: (i) increased activity at the South Pole station over
the years, which may have led to higher aerosol loads at low levels, affecting both the local environment and the field of view
of the instruments; and (ii) the influence of minor volcanic eruptions, particularly during the 2000s, when several eruptions
were known to have increased stratospheric aerosol optical depth on a global scale (Vernier et al. (2011)).

| Station | Kendall's $\tau$ | p-value | AOD change per year (%) |
|---|---|---|---|
| **Arctic** | | | |
| Barrow | -3.27e-04 | 9.96e-01 | -1.00e-02 |
| Eureka (OPAL) | 4.51e-02 | 5.65e-01 | 6.10e-01 |
| Thule | -3.39e-02 | 6.02e-01 | -4.50e-01 |
| Ny-Ålesund | -9.40e-02 | 9.85e-02 | -7.90e-01 |
| Andenes | -2.33e-01 | 4.74e-05 | -2.43 |
| **Antarctica** | | | |
| Marambio | 9.58e-02 | 3.15e-01 | 1.30 |
| Troll/Trollhaugen | 5.34e-02 | 3.38e-01 | 0.47 |
| Syowa | 3.27e-01 | 3.02e-13 | 3.84 |
| South Pole | 2.78e-01 | 5.40e-03 | 3.54 |

**Table 3.** Trend analysis for 9 polar stations, presenting Kendall's $\tau$ values as a measure of rank correlation, along with p-values to assess the
statistical significance of the observed trends.






# 6 Ship-borne measurements

| Ship-borne campaign | |
|---|---|
| **Oceanic Sectors** | **Measurement Period** |
| Eastern Chuckci Sea, Beaufort Sea and Amundsen Gulf (NAA) | March 2008 - July 2023 [85] |
| Northern Greenland - Norwegian Sea (GNS) | July 2007 - August 2024 [168] |
| Barents Sea and East Siberian Sea (BES) | April 2008 - July 2024 [27] |
| Southern Pacific Ocean (PAC) | January 2008 - December 2020 [60] |
| Antarctic Peninsula (APE) | February 2009 - February 2014 [8] |
| Southern Atlantic Ocean (ATL) | December 2007 - December 2020 [376] |
| Southern Indian Ocean (IND) | December 2007 - December 2019 [141] |

**Table 4.** List of the seven oceanic sectors defined for the ship-borne level 2.0 sun-photometer measurements conducted from 2004 to 2024. The total number of measurements is provided in parentheses.

Since 2004, several research cruises have been conducted in the Arctic and Antarctic Ocean regions, using the Microtops instrument. The instrument was calibrated at the NASA Goddard Space Flight Center (GSFC) calibration facility, following a transfer calibration procedure from a master CIMEL instrument (Smirnov et al. (2009)). Measurements focused on level 2.0 columnar aerosol radiative parameters, specifically the aerosol optical depth $\tau(0.500\mu m)$ and the Angström exponent $\alpha$. These parameters were directly downloaded from the AERONET website, considering only cruises conducted within latitudes 66°N to 90°N for the Arctic and 66°S to 90°S for Antarctica. The measurements were part of the Maritime Aerosol Network (MAN), which provides validation points for satellite observations and aerosol transport models. Details about these measurements, including the measuring points considered for this study, are available on the AERONET website (AERONET Aerosol Maritime Network: https://aeronet.gsfc.nasa.gov/new_web/maritime_aerosol_network_v3.html; last access 15/05/2025).

The Microtops II sun photometer operates in two configurations: (i) with filters centered at 0.340, 0.440, 0.675, 0.870, and 0.936 $\mu m$; or (ii) with filters centered at 0.440, 0.500, 0.675, 0.870, and 0.936 $\mu m$ (Smirnov et al. (2011)). To ensure





comparability with measurements from CIMEL instruments, only $\tau(0.500\mu m)$ and $\alpha$ values calculated for the wavelength
range 0.440 to 0.870 $\mu m$ were considered.

The Arctic cruises have been divided in three main oceanic sectors following Tomasi et al. (2015): (i) Eastern Chuckci
Sea, Beaufort Sea and Amundsen Gulf (NAA) between $170°$W and $110°$W, (ii) Northern Greenland-Norwegian Sea (GNS)
between $20°$W and $30°$E, and (iii) Barents Sea and East Siberian Sea (BES) between $30°$E and $130°$E. In Antarctica, four
different sectors have been defined: (i) Southern Pacific Ocean (PAC) between $150°$E and $75°$W, (ii) Antarctic Peninsula
(APE) between $75°$W and $50°$W, (iii) Southern Atlantic Ocean (ATL) between $50°$W and $20°$E, and (iv) Southern Indian
Ocean (IND) between $20°$E and $150°$E. Tables S1 and S2 in the supplementary material list the scientific cruises considered
for each sector and used in this study.

The ship-borne measurements were analyzed using the same criteria as those applied to the ground-based sun-photometers.
As shown in Figures 15, 16, and 17 the daily mean values of $\tau(0.500\mu m)$ and $\alpha$ were grouped into monthly subsets. These
subsets were used to calculate the monthly mean values for both parameters and to define their RFHs for all seasons.

## 6.1 Arctic Ocean

Figure 15a shows the monthly mean values of $\tau(0.500\mu m)$ and $\alpha$ derived from Microtops measurements conducted during ten
cruises in the Eastern Chukchi Sea, Beaufort Sea, and Amundsen Gulf (NAA). Although the dataset spans from March 2008 to
July 2023, only 85 observations were available between March and October. The findings align with those reported by Tomasi
et al. (2015), confirming a typical seasonal trend. The monthly mean $\tau(0.500\mu m)$ was 0.10 in March, peaked at 0.15 in April
and 0.17 in May, and gradually declined to 0.05 in September and approximately 0.04 in October. The $\alpha$ parameter remained
relatively stable from March to October, ranging between 0.90 in June and 1.50 in July. The RFH of $\tau(0.500\mu m)$ exhibited
a more dispersed distribution curve, spanning 0.02–0.30, compared to the GNS and BES sectors, particularly in spring and
summer. During summer, the mean $\tau(0.500\mu m)$, representative of background conditions, was 0.12. This value was twice
as high as that measured in the others Arctic oceanic sectors, highlighting the NAA sector as the most affected by polluted
aerosols.

A total of 168 measurements from twenty-seven scientific expeditions were used to derive the same parameters for the
Greenland Sea and Norwegian Sea (GNS) sector. Figure 15b shows that $\tau(0.500\mu m)$ values decreased from below 0.10 in
April to 0.04 in September, consistent with the behavior observed by Tomasi et al. (2015) in this region. Similar to the NAA
sector, the monthly mean $\alpha$ values in the GNS sector remained relatively stable, ranging from 1.25 in April to 1.40 in May. The
average summer values for $\tau(0.500\mu m)$ and $\alpha$ in the GNS sector, at 0.07 and 1.30 respectively, align well with those observed
at Ittoqqortoormiit (0.07 and 1.42, Fig. 5c) and Hornsund (0.06 and 1.44, Fig. 7a). These similarities occur during the summer
when polar air masses are transported toward the Svalbard Archipelago. The larger number of observations for the GNS sector
provided a clearer characterization of aerosol extinction features and challenged the reported similarity between these stations
during spring, as noted by Tomasi et al. (2015).

In the Barents Sea and East Siberian Sea (BES) sector ten cruises were conducted spanning the period from April 2008 to
July 2024, for an overall number of 27 observations. The monthly mean $\tau(0.500\mu m)$ values included a single measurement



in April and June, and values ranging from 0.03 in July to 0.06 in August. Similar to other oceanic sectors, $\alpha$ values showed no significant variations, remaining between 1.20 and 1.40. The limited number of observations for this sector prevents any meaningful statistical or climatological analysis of aerosol extinction features.





**Figure 15.** Multi-year sun-photometer measurements of aerosol optical depth $\tau(0.500\mu m)$ and exponent $\alpha$ conducted by several research institution during scientific cruises in the Arctic ocean between 2004 and 2024.





## 6.2 Antarctic Ocean

For Antarctica, the daily mean values of $\tau(0.500\mu m)$ and $\alpha$, measured using the Microtops II sun-photometer aboard scientific vessels, were categorized into four sub-sectors (Table S2): (i) Southern Pacific Ocean (PAC), (ii) Antarctic Peninsula (APE), (iii) Southern Atlantic Ocean (ATL), and (iv) Southern Indian Ocean (IND). The number of observations varied significantly between sectors, ranging from just 8 measurements in the APE sector to 376 measurements in the ATL sector. These differences should be considered when discussing the aerosol extinction features.

As shown in Table S2, thirty-nine cruises were conducted in the ATL sector from December 2007 to December 2020, collecting 376 measurements in total. The monthly mean values of $\tau(0.500\mu m)$ and $\alpha$ are presented in Fig. 16a. The AOD values remained stable from December to April, ranging between 0.02 and 0.03, which is close to the instrument's detection limit, with an average standard deviation of $\sigma = 0.01$. The monthly mean $\alpha$ values ranged from 1.20 in December to approximately 1.40 in February, with data dispersion decreasing from summer ($\sigma = 0.40$) to autumn ($\sigma = 0.33$ ). The RFH of $\tau(0.500\mu m)$ indicated AOD values between 0.01 and 0.10 across both seasons, while the $\alpha$ values showed a broader dispersion, ranging from 0.02 to 2.00.

Monthly mean values of $\tau(0.500\mu m)$ and $\alpha$ derived from 141 daily measurements performed during nineteen cruises made in the Southern Indian Ocean are shown in Fig. 16b. The monthly mean values of $\tau(0.500\mu m)$ exhibit a decreasing trend, from approximately 0.03 in December and January to 0.01 in April. The monthly mean $\alpha$ values follow a similar pattern to those in the ATL sector, ranging from about 1.40 in February to 1.00 in March. The RFH of $\tau(0.500\mu m)$ is also comparable to that of the ATL sector but with slightly less dispersion. However, the RFH of $\alpha$ shows a greater influence of larger particles, likely due to increased production of sea-salt particles.

As shown in Table 4, nine AERONET/MAN cruises were conducted in the Southern Pacific Ocean between January 2008 and December 2020, resulting in a total of 60 observations. The monthly mean values of $\tau(0.500\mu m)$ ranged from 0.04 in December to 0.09 in March, while the monthly mean $\alpha$ values decreased from approximately 1.40 in January to 1.00 in March. The RFHs indicate that almost all measurements were conducted during the summer season, with only a few taken in autumn. No observations were available for winter or spring.

Limited aerosol optical data were collected in the oceanic sector around the Antarctic Peninsula during three scientific cruises conducted between February 2009 and February 2014. As shown in Fig. 17b, only 8 measurements were recorded in this sector, all in February. The mean monthly values of $\tau(0.500\mu m)$ and $\alpha$ were 0.03 and 0.60, respectively, likely linked to the presence of sea-salt coarse particles transported from offshore areas of the Drake Passage (Posyniak and Markowicz (2009)).





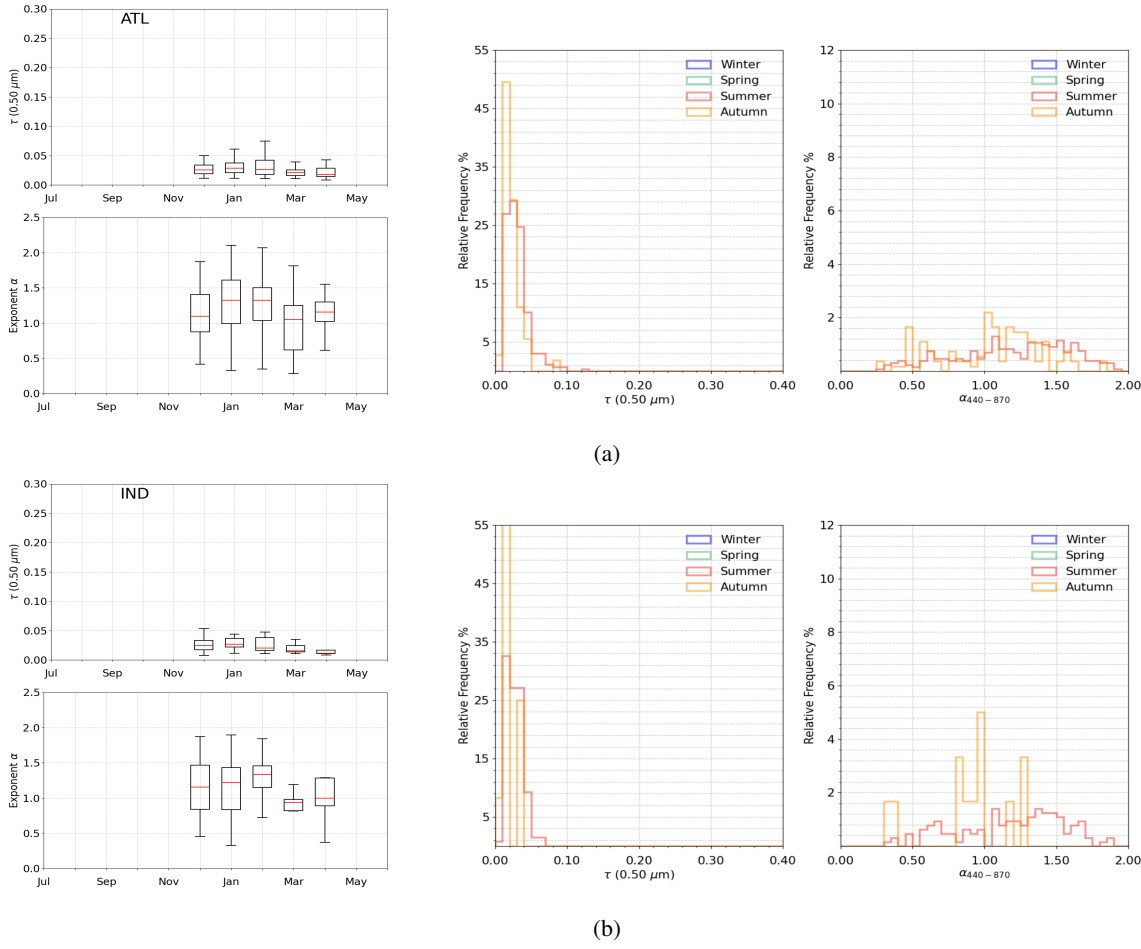

**Figure 16.** Multi-year sun-photometer measurements of aerosol optical depth $\tau(0.500\mu m)$ and exponent $\alpha$ conducted by several research institution during scientific cruises in the Antarctic ocean between 2004 and 2024.





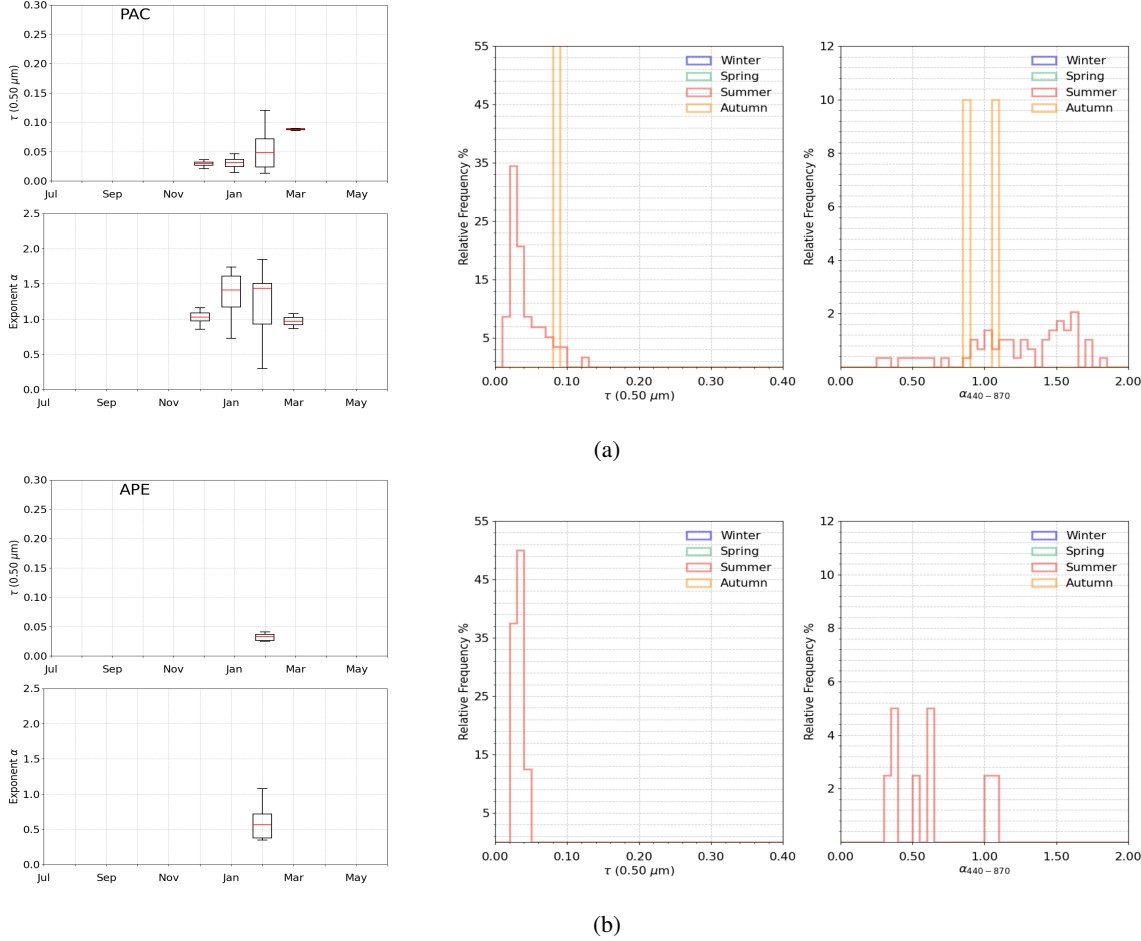

**Figure 17.** Multi-year sun-photometer measurements of aerosol optical depth $\tau(0.500\mu m)$ and exponent $\alpha$ conducted by several research institution during scientific cruises in the Antarctic ocean between 2004 and 2024.

# 7 Conclusion

Monthly mean values of $\tau(0.500\mu m)$ and the Ångström exponent $\alpha$ were determined from ground-based photometer measurements at 25 polar sites using either a sun-sky CE318 photometer or a sun-sky-lunar CE318-T photometer, both from CIMEL Electronique. Solar level 2 data and lunar level 1.5 data were downloaded directly from the AERONET website, merged into single files, and then processed. Other photometer models were also used in this study, with their characteristics described in previous sections. The results, presented by season, helped to better understand the seasonality of columnar aerosol extinction parameters and to study the evolution of these parameters over time, expanding on the work published by Tomasi et al. (2015).





A key feature of this study is the use of lunar photometry data alongside solar data. This innovative development allowed scientists to study atmospheric aerosols in polar regions even during the winter season, when there is no sunlight. This normally occurs from November to March in the Arctic and from May to October in Antarctica.

The panels in the upper part of Fig. 18 show observations conducted in the Arctic region, specifically in Alaska, Northern Canada, Greenland, Svalbard, and Northern Scandinavia. Unfortunately, due to the current global political situation, data from stations in the Russian Arctic are missing. This is a major limitation for scientists studying aerosol evolution in this region, as Russia makes up nearly 40% of the Arctic.

The analysis of measurements from the 14 Arctic sites listed in Table 1 highlights the seasonality of columnar aerosol

extinction parameters. The values of $\tau(0.500\mu m)$ were generally lower during the summer, ranging between 0.04 and 0.07, and higher during the winter, between 0.05 and 0.08. The exponent $\alpha$ also showed seasonal variation, with values ranging from 1.30 to 1.70 during summer-autumn and from 1.10 to 1.50 during winter-spring. These results align with the findings of Tomasi et al. (2015), but show slightly smaller values, indicating a cleaner atmosphere year-round, likely due to European and American regulations on anthropogenic emissions. Despite this decline, the typical seasonal aerosol behavior in the Arctic

persists, with higher $\tau$ values during winter (accompanied by lower $\alpha$ coefficients) and lower $\tau$ values in summer (with higher $\alpha$). This is due to the Arctic Haze phenomenon, caused by the transport of polluted air masses from industrialized and densely populated mid-latitude regions in Europe, Asia, and North America (Stock et al. (2014)). Between April and May, the polar dome shrinks and move to higher latitudes, isolating the North Pole's atmosphere and making it cleaner, though wildfires are becoming more frequent and intense (Dall'Osto et al. (2019); Pulimeno et al. (2024); Zielinski et al. (2020)). These effects are

evident especially at sites in North America, Canada, and Greenland (see Fig. 3, 4, and 5), where the monthly mean values of $\tau(0.500\mu m)$ show significant variability, in particular during the summer season.

In Antarctica, sun- and lunar-photometer measurements were collected at 11 stations. Of these sites, 3 are coastal sites located on the Antarctic Peninsula, 6 are located a few kilometers inland in the Atlantic sector of Antarctica, and 2 are located on the ice sheet, at almost 3000 meters of altitude.

The lower panels of Figure 18 show that median $\tau(0.500\mu m)$ values ranged from less than 0.02 on the Antarctic Plateau to no more than 0.04 at the coastal stations. Escudero, located on King George Island which hosts 11 research stations (with approximately 700 beds), represents an exception for both seasons, with median $\tau(0.500\mu m)$ values around 0.08, while Juan Carlos I had data only for the summer of 2023. The exponent $\alpha$ decreased to around 1.00–1.50 during the austral summer and autumn, indicating aerosols mainly made up of fine-mode particles. However, at Escudero and Juan Carlos I, sea-salt coarse

particles dominated the aerosol composition even in summer, probably due to the islands' large ice-free areas. During the winter-spring season, $\alpha$ values dropped to around 0.50-1.00, indicating aerosols dominated by coarse-mode particles. Notably, Vechernaya Hill station had unusually high $\alpha$ values during this period, with a median of 2.30. Compared to the findings of Tomasi et al. (2015), $\tau(0.500\mu m)$ values remained in the same range, except at the stations on the Antarctic Peninsula. However, $\alpha$ showed a negative trend, indicating an increase in coarse-mode particles influence in the atmosphere during both

seasons. In addition to the measurements analyzed in this paper from Antarctica, there are also scattered observations from




other stations, such as for example Bharati and Maitri, both managed by the National Centre for Polar and Ocean Research, India (Kannemadugu et al. (2023)).

Ship-borne sun-photometer measurements using the Microtops II instrument were conducted onboard research vessels by various international institutions between 2004 and 2024 in different sectors of the Arctic and Antarctic Oceans.

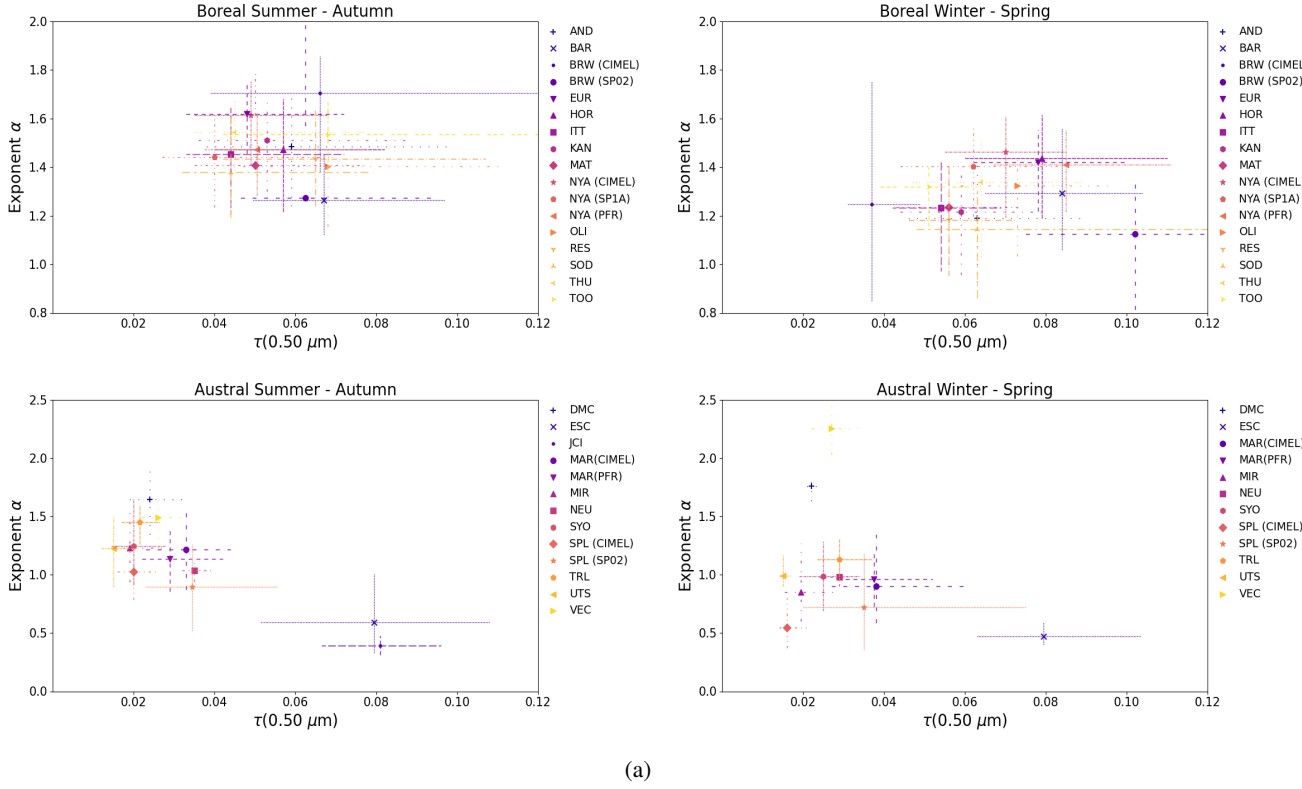

(a)

**Figure 18.** Upper part: Scatter plot shows the seasonal median values of the Ångström exponent $\alpha$ versus the corresponding seasonal median values of aerosol optical thickness $\tau(0.500\mu m)$, derived from sun- and lunar-photometer measurements at the polar sites listed in Table 1. The plot displays data for the Boreal summer-autumn and winter-spring seasons. The sites include Andenes (AND), Barentsburg (BAR), Barrow (BRW), Eureka OPAL (EUR), Hornsund (HOR), Ittoqqortoormiit (ITT), Kangerlussuaq (KAN), Matorova (MAT), Ny-Ålesund (NYA), Oliktok (OLI), Resolute Bay (RES), Sodankylå (SOD), Thule (THU), and Toolik Lake (TOO). In the upper-left panel, colored symbols represent summer–autumn results, while the upper-right panel shows winter–spring results. Vertical and horizontal dashed bars indicate the 25th and 75th percentiles. Lower part: Similar scatter plots are shown for sun- and lunar-photometer measurements at Antarctic sites listed in Table 2 during the Austral summer-autumn and winter-spring seasons. The sites include DomeC (DMC), Escudero (ESC), Juan Carlos I (JCI), Marambio (MAR), Mirny (MIR), Neumayer (NEU), Syowa (SYO), South Pole (SPL), Troll/Trollhaugen (TRL), Utsteinen (UTS), and Vechernaya Hill (VEC). The lower-left panel shows the summer–autumn results, while the lower-right panel displays the winter–spring results.





Analysis of data from forty-seven AERONET-MAN cruises in three Arctic oceanic sectors revealed that the monthly mean values of $\tau(0.50\mu m)$ ranged from 0.04 to 0.17 between March and October, as measurements were only available during periods of daylight. The higher AOD levels observed during the spring season, particularly in the NAA sector, were attributed to the intrusion of polluted air masses into the Arctic atmosphere. As expected, the typical seasonal AOD pattern observed at ground station, consisting in higher values in winter-spring and lower values in summer-autumn, is also evident in these oceanic sectors. Installing CIMEL sun-sky-lunar instruments on research vessels in the future would enable observations during the polar night, addressing the current gaps in data coverage.

In the four Antarctic oceanic sectors, aerosol extinction parameter analysis from seventy AERONET-MAN cruises conducted between 2004 and 2024 showed that monthly mean values of $\tau(0.50\mu m)$ ranged from 0.01 to 0.09. These values are approximately half the magnitude recorded in the Arctic Ocean and fall within the Microtops II instrumental uncertainty of 0.02. Due to the manual nature of these measurements and the extreme environmental conditions in Antarctica, observations were further restricted to the austral summer and autumn (December to April), when daylight permitted.

In addition to characterizing the seasonal variability of aerosol extinction parameters, this study also investigated long-term trends in $\tau(0.500\mu m)$ at selected Arctic and Antarctic AERONET sites. The trend analysis revealed a statistically significant decrease at Andenes, where aerosol optical depth declined by 2.43% per year, likely due to stricter emission regulations in Europe. In contrast, Syowa and South Pole exhibited increasing trends, with aerosol optical depth rising by 3.84% and 3.54% per year, respectively. However, the robustness of these results is limited by data availability and changes in instrumentation over time. These findings highlight the importance of continuous and standardized long-term observations to improve the understanding of aerosol trends in polar regions.

This paper aimed to understand and describe aerosol trends at both poles through a multi-year time series analysis at several stations. Although we focused mainly on AERONET sites, as CIMEL procedures are considered the most comprehensive for sun-sky-lunar measurements, other photometer models are also capable of capturing aerosol behavior in these challenging polar environments (Mazzola et al. (2024)).

This work investigates the AOD variability across polar observational networks, and in order to understand the atmospheric processes linked with aerosol distributions in the Arctic and Antarctic regions. In the Arctic, pronounced maximum seasonal AOD during winter and early spring is consistent with the well-documented Arctic haze phenomenon, attributable to enhanced transport of mid-latitude anthropogenic emissions. Stations including Barrow, Eureka, and Ny-Ålesund demonstrated these characteristic patterns, further accentuated in recent years by high AOD, low $\alpha$ cases, indicating biomass-burning aerosols associated with intensified boreal wildfire activity. In addition, Greenland stations, notably Thule and Kangerlussuaq, reported spring and summer AOD enhancements, showing the significance of Arctic haze intrusions and episodic high AOD events. Measuring sites in Scandinavia (Andenes and Sodankylä), exhibited lower seasonal variability, possibly due to the proximity to continental emissions and maritime aerosol sources. Antarctic observations revealed summer conditions dominated by smaller, fine-mode aerosols, whereas winter and spring seasons exhibited a shift toward coarse-mode particles, likely resulting from sea-spray aerosols. Positive AOD trends at Antarctic stations such as Syowa and the South Pole have been reported that further indicates potential shifts in aerosol source regions and transport pathways.





These findings show the crucial role of sustained, high-precision aerosol monitoring in polar environments, essential for our understanding of aerosol–effects in the context of the increased anthropogenic pressures to the Earths climate. Future works could expand this study by including more stations, field campaigns, and ship-borne campaigns using different photometer models at both poles. Currently, the biggest limitation is the uneven distribution of stations and measurements. In the Arctic, the Russian sector remains a black box. Since it makes up almost 40% of the Arctic, studying this area is crucial for a better

understanding of aerosol climatology. In Antarctica, most stations are located in the Antarctic Peninsula and Atlantic sector. A wider distribution of photometric instruments across other regions would help improve coverage of the entire continent.

## 8  Acknowledgments

The authors would like to acknowledge the support of the AERONET network for providing measurements from the Arctic and Antarctic stations. We also thank our colleagues from the Arctic and Antarctic Research Institute, the Zuev Institute of

Atmospheric Optics, the National Oceanic and Atmospheric Administration (NOAA), the Alfred Wegener Institute (AWI), the Physical Meteorological Observatory Davos (PMOD), and the Japan Meteorological Agency for their active involvement and participation in this study by supplying photometric measurements. Additionally, we acknowledge the P.I.s of the AERONET/-MAN cruises conducted in both the Arctic and Antarctic Oceans, during which Microtops measurements of aerosol optical thickness were collected and analyzed in this study. Funding for K.Stebel was provided by the Norwegian Environment Agency

Monitoring programme for climate gases and aerosols and MIT3D CCN-1 SVAR (ESA 4000135992/21/I-DT-lr CCN-1). Finally, we would like to thank COST Action CA21119 Harmonia, "International network for harmonisation of atmospheric aerosol retrievals from ground based photometers". This article is dedicated to Dr. Claudio Tomasi, leader of the cooperative program that in 1999 led to the creation of the Polar-AOD network, passed away in 2024.

## 9  Author contribution

Conceptualization S.P., M.M. and A.L..; data curation C.T., S.K., N.K., C.R., S.G., K.S., V.F., I.A., S.B., L.M., N.O., P.S., P.G., E.L., T.F.E., A.H., V.A., R.K., J.C., D.K., S.M.S., O.R.S., R.S.S., H.T., L.R., R.R.C., M.R., R.E., M.V.R., A.C., P.C., and J.H.; formal analysis S.P.; visualization S.P., C.F., M.M., V.V. and A.L.; writing and editing, S.P., M.M., A.L., and C.F.. All authors have read and agreed to the published version of the manuscript.

## 10  Competing interests

At least one of the (co-)authors is a member of the editorial board of Atmospheric Chemistry and Physics.





## 11 Financial support

This work was supported by COST Action CA21119 Harmonia: International network for harmonisation of atmospheric aerosol retrievals from ground based photometers, supported by COST (European Cooperation in Science and Technology).



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
