# Peer review of "Recent Advances in Aerosol Optical Depth Measurements in Polar Regions: Insights from the Polar-AOD Program"

_EGUsphere, 2025_

## Author Comment (AC1)

- **RC1**: ['Comment on egusphere-2025-2527'](), Anonymous Referee #3, 16 Sep 2025

  I have thoroughly reviewed the authors' responses to my comments and the revised manuscript. I commend the authors for their meticulous and thoughtful revisions, which have significantly strengthened the manuscript's scientific rigor, clarity, and interpretability. The responses directly address all major concerns. I am thoroughly impressed by the diligence and thoughtfulness demonstrated in addressing each point raised. The revisions significantly enhance the manuscript's clarity, methodological rigor, and scientific validity. The authors have not only provided robust justifications for their findings but also implemented precise textual refinements that markedly improve readability.

  This work makes a valuable contribution to the science community, and I recommend acceptance.

  *We sincerely thank the reviewer for his constructive feedback throughout the revision process. We are pleased that our efforts to enhance the scientific rigor and clarity of the manuscript met expectations.*

- **RC2**: ['Comment on egusphere-2025-2527'](), Anonymous Referee #2, 26 Nov 2025

  **General comments:**

  Compared with previous studies, this research offers little innovation. While it supplements some valuable polar nighttime observation data, it fails to conduct an in-depth evaluation of the data. Instead, the nocturnal results are described using speculative claims that require further verification. Additionally, the paper contains numerous contradictions. Therefore, it is not suitable for publication at this stage.

  *We thank the reviewer for its critical assessment and the time taken to provide specific feedback. We have taken these concerns seriously and have performed a revision of the manuscript to strengthen the data evaluation, clarify our interpretations, and resolve the internal contradictions noted. Below, we outline how we have addressed the three main points of concern.*

  **Specific comments:**

  1. It is crucial to conduct a thorough analysis of potential nocturnal cirrus cloud contamination rather than relying on speculation. In lines 289–290, the authors argue that the reduction in the Angstrom exponent ($\alpha$) during winter is attributable to cirrus cloud contamination in the nighttime data. However, given the low average AOD values observed in winter, the inference that this reduction is caused by cirrus cloud interference is hardly convincing.

  *While the primary focus of this study is a broad analysis of aerosol variability and long-term trends in both Polar Regions, we agree that the potential contamination of cirrus cloud in photometric measurements requires careful justification.*

*We therefore changed the sentence as follows:*

*"While the reduction in winter α values could be partially attributed to cirrus cloud contamination, this is likely not the sole driver. First, given that AOD levels in December are extremely low (< 0.05), the accuracy of the Ångström Exponent is inherently limited. Similarly, if the low α values were caused exclusively by thin cirrus, corresponding high AOD values would be expected, which is not observed in our data."*

*We also added this general sentence in the introduction:*

*"In Polar Regions, thin cirrus clouds are frequent in the upper troposphere and lower stratosphere, often above aerosol layers (Engelmann et al., 2021). Regarding the potential of this contamination, it is well-established in literature that when cirrus is not detected by cloud-screening, its optical depth is misinterpreted, creating a bias in AOD. From satellite, near detected cirrus form a zone where pixels flagged as clear still show enhanced reflectance and AOD. Since cirrus consist of large ice crystals, their contribution in nominally clear pixels produces higher AOD and lower AE (Wollner et al., 2014, O'Neill et al., 2016)."*

2. As is well known, aerosols generated by wildfires (i.e., biomass-burning aerosols, BBA) are dominated by fine-mode particles, with an Angstrom exponent (α) typically greater than 1.5—a point the authors have repeatedly referenced in the main text. Yet why does the authors argue in the conclusion that the increasing prevalence of cases characterized by high AOD and low α is associated with intensified boreal wildfire activity?

*Thank you for pointing out this inconsistency. The statement in the conclusion mentioning low AE in association with wildfire activity was indeed a typographical error. Our data for extreme wildfire events, such as those discussed in Chapter 4 and 6, correctly show an increase in AOD accompanied by a shift toward higher alpha values during these episodes. We have corrected the text in the conclusion to reflect this; we apologize for the confusion caused by this oversight.*

3. Similarly, anthropogenic aerosols are also dominated by fine-mode particles. Why, then, does the author argue that Arctic persistent haze events associated with anthropogenic emissions should exhibit a low Angstrom exponent (α)?

*We appreciate this insightful comment regarding the relationship between anthropogenic haze and α. It is correct that anthropogenic pollutants are fine-mode particles and, in isolation, would contribute to a higher α. However, the typical seasonal aerosol behaviour mentioned in the manuscript refers to the bulk optical properties of Arctic winter/spring atmosphere as measured by photometry. As discussed in Section 6.4, the lower α values recorded during the Arctic Haze season result from several factors that overlap with anthropogenic plumes: aerosol mixing and potential cloud contamination. We have revised the manuscript to clarify that while anthropogenic components are fine-mode, the overall reduction in α during winter reflects a complex mixture of pollution and natural coarse particles.*

*"Despite this decline, the typical seasonal aerosol behaviour in the Arctic persists, with higher τ values during winter and spring. While anthropogenic Arctic Haze is primarily*

*composed of fine-mode aerosols from industrialized and densely populated mid-latitude regions in Europe, Asia, and North America (Stock et al., 2014), the overall columnar α values observed are often lower. This is likely due to the mixing of pollution plumes with natural coarse-mode particles such as sea-salt."*

4. Some of the causal analyses in the paper are quite confusing. For instance, in lines 241–244, the authors argue that pronounced right-skewed tail in winter may be attributed to higher AOD values in March and April. However, the authors define winter as December–February, so why would the higher AOD values in March and April contribute to the long tails of the winter histograms?

*Thank you for pointing out this inconsistency. This oversight occurred because the seasonal analysis was originally divided into only two periods (summer-autumn and winter-spring), but was later split into the four standard seasons. We have revised the manuscript to clarify that the pronounced right-skewed tails in the winter histograms (December–February) are primarily driven by early Arctic Haze intrusions occurring in February. The peak AOD values observed in March and April have been correctly reattributed to the spring season. We have updated the text in Section 6 to ensure full consistency between our seasonal definitions and the causal analysis of the relative frequency histograms.*
*"The long-tail feature observed during winter is attributed to early Arctic Haze intrusions occurring in February, while the spring tails are driven by the peak of the phenomenon in March and April."*

5. The paper argues that persistent haze events in the polar winter give rise to peak AOD, while the atmosphere is generally cleaner in summer (e.g., Lines 217-218). However, in terms of observed AOD values, summer AOD is considerably higher than that in winter (Lines 239-241) —it seems that the conclusion is inconsistent with observational facts.

*Please note that lines 217-218 describe the values obtained at Barrow, while lines 239-241 at Toolik Lake. In general, we understand the point, but the inconsistency is only apparent if you look at the seasonal distribution. We added this paragraph to better explain.*

*"The seasonal AOD distributions at Barrow and Toolik Lake exhibit characteristics that are consistent with other Arctic observatories. The distributions are notably shifted toward higher values during the spring—and occasionally winter—reflecting the presence of the Arctic Haze. Conversely, summer and autumn exhibit lower median AOD values; however, these two seasons are characterized by elongated distribution tails extending toward higher values (which justify the higher mean values). Regarding the Ångström Exponent α, the seasonal behavior shows a distinct divergence: the winter distribution is skewed toward lower values, suggesting a larger effective particle size. In contrast, the summer distribution is shifted toward higher α values, consistent with a dominance of fine-mode aerosols. The distributions for spring and autumn remain relatively similar to one another, representing transitional states in the aerosol regime."*

6. Instrumental biases are non-negligible, as evidenced by substantial discrepancies in the Angstrom exponent (α) — particularly in December — between CIMEL and PFR measurements at the Marambio station. The manuscript lacks systematic

documentation of calibration methodologies across different stations and instruments. It should explicitly specify which stations adopt standard transfer calibration as opposed to Langley method calibration. For stations utilizing Langley calibration, additional quality control measures ought to be implemented. For reference, Che et al. (2025) have documented several significant advancements in techniques for modifying and improving Langley calibration.

*We agree that calibration methodologies are of fundamental importance, especially in photometric studies conducted in remote environments like Polar Regions, where the background aerosol load is extremely low. Similarly, as stated in the text, different quality assurance methods applied by AERONET for the CIMEL and by PMOD for the PFR could partially justify the difference in retrieved AOD and, consequently, α values. However, the primary goal of this manuscript is to provide a comprehensive climatological analysis across a vast network of stations using various instrument models and analysis procedures. To ensure data integrity, we requested and utilized only the highest-quality, quality-assured data available from each contributing research group and network. Including a detailed technical documentation of specific methodologies (e.g., distinguishing between standard transfer calibration and the Langley method) for every single station and instrument would have shifted the focus away from the main scientific objectives of the study.*

*Nevertheless, in order to acknowledge the point raised by the reviewer, we modified the text as follows:*

*"At Marambio, the monthly AOD time series shows qualitative consistency between the CIMEL and PFR instruments, with both capturing a seasonal cycle defined by maximum values in late autumn/winter and minima in summer. Conversely, the Ångström Exponent α distributions exhibit notable discrepancies. These differences likely stem from the distinct calibration chains and quality assurance protocols maintained by AERONET (for the CIMEL) and PMOD (for the PFR). Regardless, it is critical to emphasize that the accuracy of α is fundamentally constrained by the AOD magnitude, with uncertainties increasing significantly as AOD decreases. Furthermore, as stated above, the winter PFR measurements rely on sun photometry—often performed at high solar zenith angles—whereas the CIMEL utilizes lunar photometry. This difference in radiation sources and the resulting signal-to-noise ratios contributes further to the observed divergence in the α distributions."*

7. Figure 1 would benefit from the inclusion of geographic coordinates (latitude and longitude).

*We have carefully considered the inclusion of geographic coordinates directly on the maps. However, we have decided to keep the figure in its current form for the following reasons: (i) Readability: The maps already contain a high density of information, including identification numbers for 15 Arctic and 11 Antarctic sites. Adding latitude and longitude grids or labels directly onto the images would, in our view, significantly clutter the visual presentation and reduce the legibility of the station locations. (ii) Data Availability: To ensure the reader has access to precise location data, the exact geographical coordinates (latitude and longitude) and altitudes for every station analyzed are explicitly detailed in Table 6.1 for the Arctic and Table 6.2 for Antarctica. We believe that providing this information in tabular form is the most effective way to maintain a clean visual overview in the figure while ensuring scientific precision.*

8. Incorporate a multi-year mean AOD spatial distribution map to better visualize geographic variability patterns.

*We appreciate the suggestion to include a spatial distribution map; however, we have respectfully opted not to include one for several methodological reasons:*
- *the strength and focus of this study lie in the ground-based measurements collected at specific, isolated stations. Given the vast distances between these sites in the Arctic/Antarctic regions, interpolating these data points to create a continuous spatial map would be mathematically speculative. Such a map would likely generate artifacts that do not represent the true atmospheric state, potentially misleading the reader regarding regional gradients. Unlike satellite-derived products (e.g., MODIS), which provide the pixel density required for spatial mapping, our station-based data are 'point measurements'.*
- *the primary objective of this paper is to characterize the annual and seasonal evolution of aerosols, particularly during the challenging transition from the polar night to spring and from autumn back to polar night. A static spatial map would not capture this dynamic temporal variability, which we believe is more effectively communicated through the current time-series and distribution plots.*

*In summary, we feel that maintaining the focus on site-specific temporal analysis ensures a higher level of scientific rigor and avoids the uncertainties inherent in interpolating sparse polar data.*

9. Provide thorough discussion of outliers in Figure 14a.

*We added the following text to the paper.*

*"For the sake of completeness, the four highest AOD peaks observed at Andenes in the time series are attributable to well-documented long-range transport episodes. The peak in July 2015 is consistent with the arrival of intense North American boreal wildfire smoke in the Arctic, as reported by Markowicz et al. (2016), who documented widespread elevated AOD over northern Europe and the Arctic during this period. The peaks observed in 2019 (spring and late summer) are linked to exceptional aerosol loading from biomass-burning plumes combined with volcanic sulfate following the Raikoke eruption, as demonstrated by Herrero-Anta et al. (2025), who showed persistent elevated aerosol layers affecting Arctic and sub-Arctic sites. Finally, the extreme AOD peak in May 2023 coincides with the onset of the unprecedented Canadian wildfire season, during which satellite observations reported massive smoke transport toward the Arctic and northern Europe (Copernicus, 2023). Together, these events explain the episodic high-AOD outliers superimposed on the long-term declining trend at Andenes."*

**Technical corrections:**

1. Line 688, replace 'move' with 'moves'.

*Amended*

2. The manuscript exhibits a significant number of in-text citation formatting errors, e.g., line 27, '(Klonecki et al. (2003))' should be '(Klonecki et al., 2003)'

*We went through the paper and checked the formatting errors.*